# Interplay between charge distribution and DNA in shaping HP1 paralog phase separation and localization

**Tien M Phan[1], Young C Kim[2], Galia T Debelouchina[3]\*, Jeetain Mittal[1,4,5]\***

[1]Artie McFerrin Department of Chemical Engineering, Texas A&M University, College Station, United States; [2]Center for Materials Physics and Technology, Naval Research Laboratory, Washington, United States; [3]Department of Chemistry and Biochemistry, University of California, San Diego, La Jolla, United States; [4]Department of Chemistry, Texas A&M University, College Station, United States; [5]Interdisciplinary Graduate Program in Genetics and Genomics, Texas A&M University, College Station, United States

**\*For correspondence:** gdebelouchina@ucsd.edu (GTD); jeetain@tamu.edu (JM)

**Competing interest:** The authors declare that no competing interests exist.

**Preprint posted** 18 July 2023

**Sent for Review** 18 July 2023

**Reviewed preprint posted** 05 October 2023

**Reviewed preprint revised** 08 February 2024

**Version of Record published** 09 April 2024

**Abstract** The heterochromatin protein 1 (HP1) family is a crucial component of heterochromatin with diverse functions in gene regulation, cell cycle control, and cell differentiation. In humans, there are three paralogs, HP1α, HP1β, and HP1γ, which exhibit remarkable similarities in their domain architecture and sequence properties. Nevertheless, these paralogs display distinct behaviors in liquid-liquid phase separation (LLPS), a process linked to heterochromatin formation. Here, we employ a coarse-grained simulation framework to uncover the sequence features responsible for the observed differences in LLPS. We highlight the significance of the net charge and charge patterning along the sequence in governing paralog LLPS propensities. We also show that both highly conserved folded and less-conserved disordered domains contribute to the observed differences. Furthermore, we explore the potential co-localization of different HP1 paralogs in multicomponent assemblies and the impact of DNA on this process. Importantly, our study reveals that DNA can significantly reshape the stability of a minimal condensate formed by HP1 paralogs due to competitive interactions of HP1α with HP1β and HP1γ versus DNA. In conclusion, our work highlights the physicochemical nature of interactions that govern the distinct phase-separation behaviors of HP1 paralogs and provides a molecular framework for understanding their role in chromatin organization.

## eLife assessment

This **valuable** study substantially advances our understanding of molecular mechanisms driving the phase separation behavior of HP1 paralogs. The evidence supporting the conclusions is **convincing**, with rigorous and well-designed computational simulations. The work will be of broad interest to biophysicists and biochemists.

## Introduction

The HP1 family are evolutionarily conserved nuclear proteins that are essential regulators of chromatin structure and function in eukaryotic cells. In humans, the HP1 family comprises three paralogs, HP1α, HP1β, and HP1γ, encoded by a class of genes known as the chromobox (CBX) genes (CBX5, CBX1, and CBX3, respectively). All three paralogs localize to heterochromatic regions where they mediate chromatin condensation and gene silencing (*Zeng et al., 2010*; *Larson and Narlikar, 2018*). However, HP1β and HP1γ have also been found in euchromatic regions that express active genes (*Eissenberg*

*and Elgin, 2000*; *Hediger and Gasser, 2006*; *Kwon and Workman, 2011*). The importance of HP1 proteins in genome regulation and their potential association with cancer development (*Jeon et al., 2022*) has prompted increasing endeavors to elucidate the molecular mechanisms underlying their biological activity.

HP1 paralogs are multi-domain proteins, sharing high similarities in amino acid sequence and domain architecture. They possess a basic structure that consists of two highly conserved folded domains, the chromodomain (CD) and the chromoshadow domain (CSD), connected by a disordered hinge of variable lengths and two unstructured N and C terminal extensions (NTE and CTE, respectively). The CD specifically recognizes methylated lysine 9 on the histone H3 tail (H3K9me), an epigenetic mark for gene silencing (*Lachner et al., 2001*; *Bannister et al., 2001*). The CSD mediates the dimerization of HP1 and interacts with other proteins through a PXVXL motif (*Cowieson et al., 2000*; *Smothers and Henikoff, 2000*; *Thiru et al., 2004*). Dissociation constant ($K_d$) values in the nano-molar range have been reported for CSD homodimerization of HP1α and HP1β (*Her et al., 2022*; *Brasher et al., 2000*). Moreover, in vitro and in vivo co-immunoprecipitation have shown that mammalian HP1 paralogs can form heterodimers directly with one another (*Nielsen et al., 2001*), which raises the question of the biological functions of these heterodimers. Compared to the CD and CSD, the flexible disordered regions are less conserved, potentially contributing to the distinctive functional properties of different HP1 paralogs. The hinge region has been observed to target HP1 to heterochromatic regions through non-specific binding to DNA and RNA (*Keenen et al., 2021*; *Muchardt et al., 2002*). Furthermore, interactions between the hinge region and the CTE are proposed to be responsible for an auto-inhibited dimer conformation (*Larson et al., 2017*). By contrast, the NTE has been implicated in mediating the binding affinity of the CD for the methylated histone H3 tail (*Canzio et al., 2014*). Different HP1 domains contribute to a complex interaction network with specific regions of chromatin, other peptide ligands, and nuclear components. These interactions are attributed to the multi-functionality of HP1 proteins in the context of heterochromatin.

Heterochromatin formation has been suggested to involve the recruitment of HP1 paralogs based on direct binding to H3K9me3, which facilitates the construction of bridges between nucleosomes and potentially induces the formation of compact chromatin states (*Hiragami-Hamada et al., 2016*; *Kilic et al., 2018*; *Machida et al., 2018*; *Erdel et al., 2020*). However, in vitro experiments have shown that HP1α can undergo LLPS upon either NTE phosphorylation (pHP1α) or DNA binding, which provides a new perspective on heterochromatin formation and function through a phase-separation mechanism (*Keenen et al., 2021*; *Larson et al., 2017*). Despite phylogenetic conservation, HP1 paralogs display different phase separation behaviors. Without phosphorylation or DNA binding, HP1α can undergo LLPS at high protein concentrations (*Keenen et al., 2021*; *Qin et al., 2021*) and low salt concentrations (*Wang et al., 2019*). While HP1γ undergoes LLPS at a protein concentration that is multi-fold higher than the threshold concentration for HP1α (*Qin et al., 2021*), HP1β does not form liquid droplets under any tested conditions (*Larson et al., 2017*; *Qin et al., 2021*). However, HP1β, but not HP1α or HP1γ, can form liquid-like droplets in the presence of histone proteins (*Qin et al., 2021*). These observations raise fundamental questions regarding the sequence features of HP1 paralogs. For example, what is the physicochemical nature of interactions that dictate the distinct phase-separation behaviors of HP1 paralogs? How does the differential phase separation behavior enable the co-localization of HP1 proteins within heterochromatic regions, and does heterodimerization play a role in regulating this process? Do HP1 paralogs play distinct, cooperative, or largely redundant roles in heterochromatin organization? It is essential to uncover the underlying forces driving HP1 LLPS to understand the mechanisms of heterochromatin formation and regulation.

Addressing the challenges of timescales and length scales inherent in protein LLPS, a recent coarse-grain (CG) framework (*Dignon et al., 2018b*; *Regy et al., 2021b*; *Szała-Mendyk et al., 2023*) has emerged as a powerful computational tool to provide insights into the molecular interactions behind phase separation of multidomain and intrinsically disordered proteins (IDPs). Here, we use this CG simulation framework to uncover the underlying forces driving phase separation of HP1 paralogs. We examine the intricate interplay between folded and disordered domains in mediating phase separation of HP1 homo- and heterodimers. We further explore the potential co-localization of different HP1 paralogs in multicomponent assemblies and the essential role of DNA in modulating this process. Our findings reveal that strong HP1α-DNA binding outcompetes other possible heterotypic interactions between paralogs and paralogs and DNA, orchestrating the organization of the multicomponent

condensates. This study presents a comprehensive molecular framework that elucidates the mechanism and regulation of HP1 LLPS, thereby providing invaluable insights into the formation of heterochromatin and its diverse functional implications.

## Results

### Amino acid composition and distribution of HP1 paralogs

HP1 proteins consist of highly conserved folded domains and less conserved disordered regions (*Figure 1a*). Compared to the corresponding regions of HP1α, the NTE, hinge, and CTE of HP1β share 35%, 33%, and 38% sequence similarity, respectively, while for HP1γ those numbers are 15%, 36%, and 19%, respectively (*Canzio et al., 2014*). On the other hand, the CD and CSD domains of HP1α and HP1β show over 80% sequence similarity, while the similarities between these domains in HP1α and HP1γ are 71% and 87%, respectively (*Canzio et al., 2014*). Three-dimensional (3D) structures of the CD and CSD domains have been solved by nuclear magnetic resonance (NMR) spectroscopy (*Brasher et al., 2000*; *Nielsen et al., 2002*; *Richart et al., 2012*) and X-ray crystallography (*Kaustov et al., 2011*; *Kang et al., 2011*). The CD and CSD are both globular domains that share remarkable similarities. They consist of three anti-parallel β-strands at the N-terminus and an α-helix at the C-terminus. The structural difference between the monomeric units of CD and CSD can be attributed to the presence of an additional α-helix in the CSD (*Figure 1b*). It should be noted that the CD functions as a monomer while the CSD forms a symmetrical homodimer where the two CSD monomers interact via their α-helices (*Figure 1c*). Furthermore, in the crystal structure (3i3c) (*Edwards and Arrowsmith, 2009*), an additional CSD-CSD interaction can be observed, which occurs at the hydrophobic β-sheet interface of HP1α (*Figure 1c*). This interface has been observed only in this crystal structure so far, and it is not clear if it plays a role in HP1α oligomerization in solution or under physiological conditions.

HP1 paralogs are enriched in charged residues, with Arg, Lys, Glu, and Asp comprising 39–45% of the total number of amino acids in the protein. Of the three paralogs, HP1β has the highest percentage of charged residues (45%), with a predominance of negatively charged amino acids (26%). This results in a net charge of –13, which is more negative than the net charge of HP1γ (–5) and HP1α (–3). Ion-exchange chromatography also showed that HP1β has the lowest isoelectric point (pI 4.85), followed by HP1γ (pI 5.13) and HP1α (pI 5.71) (*Qin et al., 2021*). The three paralogs share comparable composition of other amino acid types such as aromatic residues (~8%), aliphatic residues (21–25%), polar residues (~15–18%), and other residues (9–11%) (*Figure 1d*).

As the sequences of HP1 paralogs contain a high fraction of charged residues, we first characterized the charge property differences at the domain level across the three paralogs. Analysis of the charge distribution on folded domains of HP1 paralogs revealed a slight variation in their net charge. Particularly, HP1α has a smaller net negative charge on its CD and exhibits a CSD with a more negative charge compared to those of HP1β and HP1γ (*Figure 1e and f*). Disordered regions are less conserved and contain the most variable amino-acid sequences. The NTE exhibits variable length and a different fraction of oppositely charged residues organized into a charge-segregated diblock (*Figure 1g–i*). HP1β has a longer negative block than HP1α, resulting in a higher net negative charge. In contrast, HP1γ has an extended positive block, leading to a net positive charge (*Figure 1e*). It is important to note that the HP1α NTE has a unique block of four serine residues (*Figure 1a*) that are constitutively phosphorylated in vivo (*Hiragami-Hamada et al., 2011*). Phosphorylation can dramatically change the charge on the HP1α NTE and significantly reduce the minimum concentration required for LLPS in the absence of DNA (*Her et al., 2022*; *Larson et al., 2017*; *Hiragami-Hamada et al., 2011*; *Nishibuchi et al., 2014*).

On the other hand, the CTE of all HP1 paralogs is characterized by a net negative charge. The charged residues in the CTE of HP1β are organized into alternating blocks of opposite charges, whereas they are relatively mixed in the CTE of HP1α. Notably, HP1γ has a considerably shorter CTE comprising only negatively charged residues. The hinge region is particularly important for determining the phase separation properties of HP1 proteins. All three paralogs have conserved basic patches located in the middle and near the end of the hinge, but HP1α has an additional basic patch at its beginning (*Figure 1g–i*). Although HP1β contains a similar number of charged residues within its hinge as HP1α, it is relatively balanced between oppositely charged residues (13/12). Conversely,

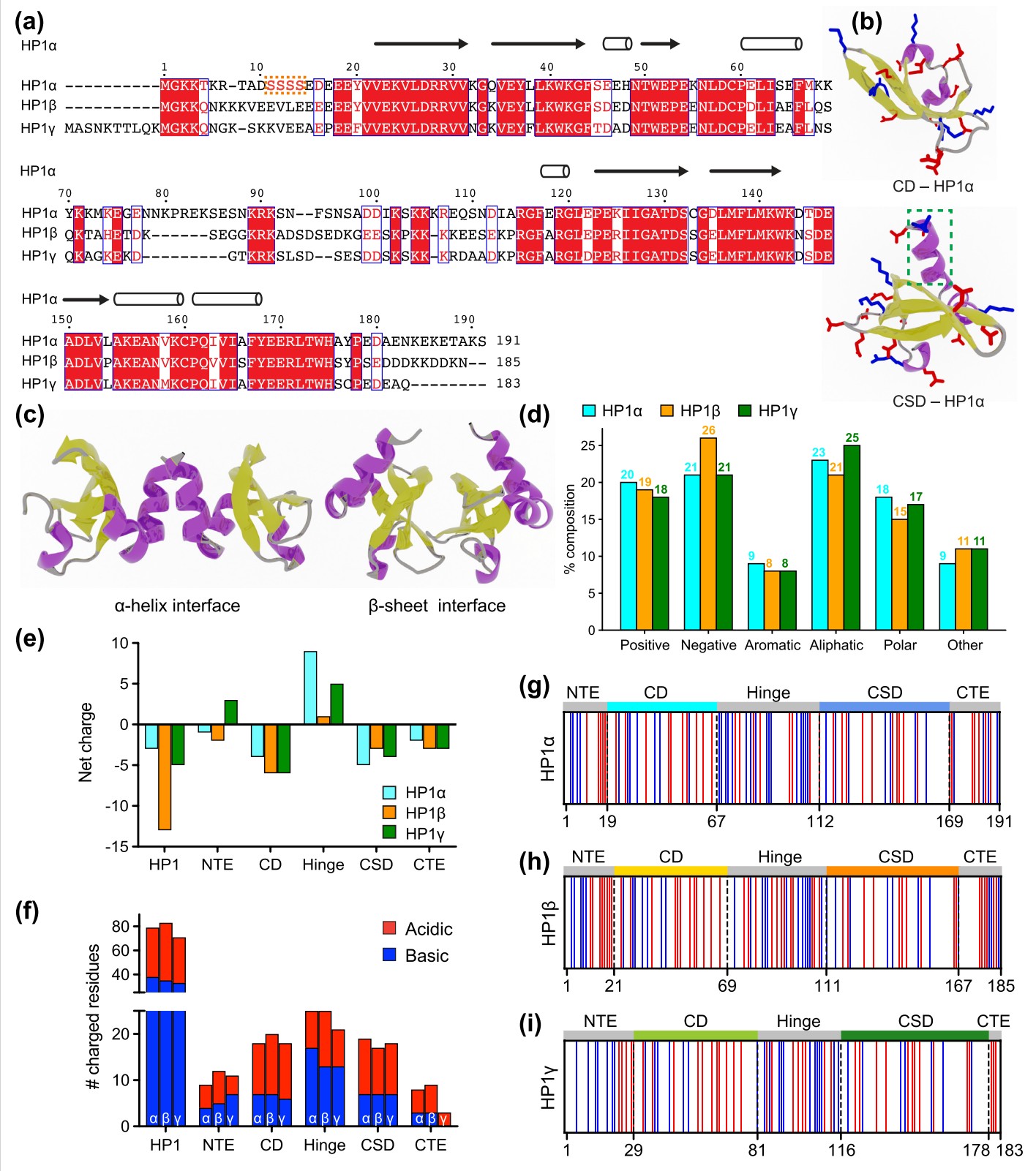

**Figure 1.** Properties of human heterochromatin protein 1 (HP1) paralogs. (**a**) Multiple sequence alignment of human HP1α, HP1β, and HP1γ. Red boxes with white letters show identical amino acids, while white boxes with red letters indicate amino acids with similar properties. The orange box highlights the unique block of four serine residues in HP1α NTE that can be constitutively phosphorylated in vivo. (**b**) Structures of the HP1α chromo (CD – PDB code 3fdt) and chromoshadow (CSD – PDB code 3i3c) domains. Positively and negatively charged residues are shown in blue and red licorice

*Figure 1 continued on next page*

Figure 1 continued

representations, respectively. The green dashed box marks the additional helix in the chromoshadow domain (CSD) compared to the chromodomain (CD). (**c**) HP1α CSD-CSD homodimer can dimerize via α-helix and β-sheet binding interfaces. (**d**) Amino acid composition of HP1 paralogs is organized into Positive (Arg, Lys), Negative (Asp, Glu), Aromatic (His, Phe, Tyr, Trp), Aliphatic (Ala, Ile, Leu, Met, Val), Polar (Asn, Gln, Ser, Thr), and Other (Cys, Gly, Pro). (**e, f**) Differences in the net charge and charged amino acids content of each domain of the HP1 paralogs. (**g, h, i**) Charge distribution in the sequences of the HP1 paralogs (blue = positively charged residues, red = negatively charged residues).

HP1α and HP1γ contain more positively than negatively charged residues in their hinge regions (17/8 and 13/8, respectively).

Our sequence comparison shows that HP1 paralogs share a common structure but differ in net charge and charge distribution along the sequence. HP1β has the most net negative charge with balanced oppositely charged residues in its hinge. HP1γ has a significantly shorter CTE but possesses an extended charged-segregated diblock in the NTE. HP1α has the least net negative charge of the three paralogs, and its hinge is rich in clusters of positively charged lysine and arginine residues. The sequence variances may dictate differences in the overall conformation and phase separation behavior of the three paralogs, as discussed below.

## Sequence variation of HP1 paralogs leads to differential conformation and phase separation

*Sequence variation affects the conformations of HP1 paralogs in the dilute phase.* We performed CG simulations of single homodimers using the HPS-Urry model (*Regy et al., 2021a*), which we recently applied to uncover the molecular interactions driving the phase separation of HP1α (*Her et al., 2022*). The full-length HP1α model was constructed as previously described using PDB structural models 3fdt (*Kaustov et al., 2011*) and 3i3c (*Edwards and Arrowsmith, 2009*) for the CD and CSD domains, respectively. Intrinsically disordered regions (IDRs), NTE, hinge, and CTE, were connected to the folded domains, while the homodimer CSD-CSD α-helix interface was created using the MODELLER software (*Eswar et al., 2007*). The HP1α dimer was used as a template to prepare HP1β and HP1γ dimers with homology modeling in MODELLER. To mimic the movements of the dimer configuration, we used the simulation protocol described previously (*Her et al., 2022*), in which the folded domains (CD and CSD) were kept rigid to avoid unfolding, the CSD-CSD domains were fixed with respect to each other, and the IDRs remained flexible.

To gain insights into the global conformation of HP1 paralogs (*Figure 2—video 1*), we calculated the radius of gyration ($R_g$) distributions of the homodimers under dilute conditions (*Figure 2a*). We found that the HP1β conformations ($R_g$ = 3.93 ± 0.78) are on average more extended than the HP1α and HP1γ structures ($R_g$ = 3.20 ± 0.31 nm and $R_g$ = 3.48 ± 0.69 nm, respectively), which is consistent with observations in previous work (*Larson et al., 2017*; *Munari et al., 2013*; *Latham and Zhang, 2021*). Notably, the average dimension of HP1 paralogs expands in order of increasing the net negative charge of the respective homodimer where HP1α has a negative charge q = –6, HP1γ has q = –10, and HP1β has q = –26. To characterize molecular interactions within the homodimer that cause the observed conformational differences, we computed the number of intramolecular (within an HP1 monomer) and intermolecular (between two HP1 monomers) van der Waals (vdW) contacts formed by each residue in the CG simulations. The contacts within the rigid body (folded domain) were excluded from the calculations. As the HP1 paralogs share similar structural topologies, we summed over the time-averaged intra- and intermolecular contact pairs per residue in each domain to create one-dimensional (1D) contact maps (*Figure 2b*). These 1D contact maps highlight the differences in overall intra- and intermolecular contacts contributed by each domain to the interaction network of HP1 homodimers. Among the three paralogs, HP1β has the smallest number of contacts at the domain level, presumably due to charge repulsion between extended stretches of acidic residues in its domains. While HP1α and HP1γ make similar contacts with their folded domains, the hinge and CTE of HP1α are much more interaction-prone in both monomer and dimer contexts.

To gain more insights into the interactions between domains, we analyzed two-dimensional (2D) time-averaged contact maps of interactions as a function residue number (*Figure 2c–e*). These maps display the high and low-contact-prone regions along the homodimer sequence. To highlight the distinct molecular interaction differences between the HP1 homodimers, we performed pairwise comparisons of the 2D contact maps. Similar to the 1D interaction map, the interaction network

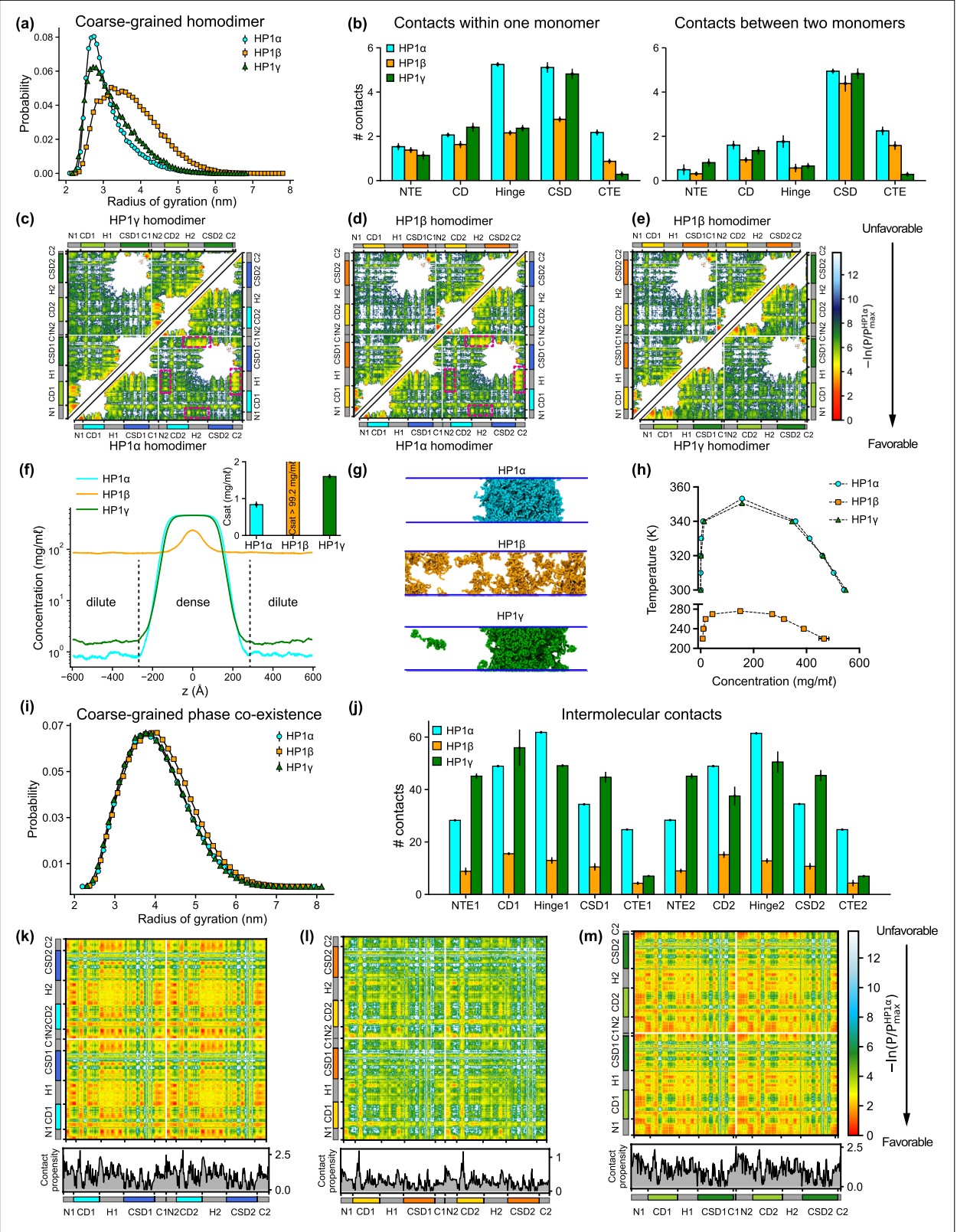

**Figure 2.** Sequence differences result in differences in conformation and phase separation of heterochromatin protein 1 (HP1) paralogs. (**a**) $R_g$ distributions of HP1 paralogs in coarse-grain (CG) single homodimer simulations using the HPS-Urry model. (**b**) Average intramolecular contacts within one chain and average intermolecular contacts between two chains on each domain of the three HP1 paralogs. Two residues were considered to be in contact if the distance between them was less than 1.5 of their vdW arithmetic mean. (**c, d, e**) Pairwise two-dimensional (2D) contact maps between

*Figure 2 continued on next page*

*Figure 2 continued*

two HP1 homodimers. The intramolecular interactions within one monomer and the intermolecular interactions between two monomers are shown in the two small triangles and the off-diagonal quadrant, respectively. The contact propensity of HP1β and HP1γ is normalized to the highest contact propensity of HP1α. The magenta boxes highlight strong electrostatic attractions between the hinge and C terminal extension (CTE) regions in HP1α. (**f**) Density profiles of HP1 paralogs in CG coexistence simulations. The inset shows the respective saturation concentrations. (**g**) Representative snapshots of each system in the CG coexistence simulations. (**h**) Phase diagrams of HP1 paralogs at different temperatures calculated in CG simulations where a dilute phase of free monomers co-exists with a dense phase. The critical temperatures of HP1α, HP1β, and HP1γ are 353.3 K, 276.1 K, and 350.5 K, respectively. (**i**) $R_g$ distributions of the HP1 paralogs as calculated in the CG phase coexistence simulations. (**j**) Average intermolecular contacts made by each domain of the HP1 paralogs in CG phase coexistence simulations. The error bars represent the standard deviation from triplicate simulation sets. (**k, l, m**) Intermolecular contacts within the condensed phase of HP1 paralogs along the sequence of each homodimer. The panels below each map show the average contacts per chain as a function of residue number. The contact propensity of HP1β and HP1γ is normalized to the highest contact propensity of HP1α. (**a-e**) show the results obtained under dilute conditions, while (**f–m**) illustrates the conditions of phase coexistence. The CG simulations under dilute and phase coexistence conditions were conducted using the HPS-Urry model at 320 K and 100 mM salt concentration.

The online version of this article includes the following video and figure supplement(s) for figure 2:

**Figure supplement 1.** AA and CG simulations of heterochromatin protein 1 (HP1) paralogs.

**Figure supplement 2.** Salt-dependent effects on the liquid-liquid phase separation (LLPS) of HP1α homodimers.

**Figure supplement 3.** Finite-size analysis.

**Figure 2—video 1.** Coarse-grain (CG) simulations of single-chain heterochromatin protein 1 (HP1) paralogs.
https://elifesciences.org/articles/90820/figures#fig2video1

**Figure 2—video 2.** Coarse-grain (CG) phase coexistence simulations of heterochromatin protein 1 (HP1) paralogs.
https://elifesciences.org/articles/90820/figures#fig2video2

of HP1α has significant contributions from the hinge and the CTE (*Figure 2c*). These interactions stem from intra- and intermolecular contacts, primarily between the CTE and the hinge and partially between the NTE and the hinge (magenta boxes). This can be attributed to the charge attraction between oppositely charged residues in the two regions. We also observe intramolecular and intermolecular interactions between the folded CD and CSD domains with disordered segments (NTE and CTE) in HP1α. HP1γ shares a similar interaction pattern between folded domains and disordered regions. However, the extended stretch of basic residues in the diblock of the HP1γ NTE promotes self-interactions within the diblock, as well as intramolecular interactions between the NTE and the CD, and intermolecular interactions between NTEs (NTE1-NTE2). The interactions in the hinge of HP1γ are less pronounced than those observed in HP1α, likely due to the absence of an additional attraction hotspot. The most significant difference between the contact maps of the paralogs is the limited and sparse interaction network between HP1β monomers (*Figure 2d and e*). The lack of favorable intermolecular interactions between disordered and folded domains causes the extended conformation of HP1β. The results of our CG simulations are in good agreement with those of atomistic simulations performed over a 5 μs simulation time (*Figure 2—figure supplement 1a–h*). Although HP1 paralogs share remarkable similarities in the sequence and structural topology of folded domains, our contact analysis suggests that sequence variations within disordered regions may cause distinct intra- and intermonomer interactions, leading to conformational differences.

## Sequence variance in HP1 paralogs leads to their distinct phase separation behaviors

We conducted CG phase coexistence simulations of the HP1 homodimers using slab geometry (*Blas et al., 2008*; *Silmore et al., 2017*), as done in our previous work (*Her et al., 2022*; *Dignon et al., 2018b*). We simulated the systems at 320 K and plotted the protein densities as a function of the z-coordinate that separated the dense phase from the dilute phase (*Figure 2f*). As HP1 can form a homodimer via the α-helix or β-sheet binding interface of the CSD (*Figure 2—figure supplement 1j*), we tested the phase separation propensity of these two configurations. In the CG coexistence phase simulations, the CSD-CSD domains were fixed with respect to each other as a single rigid body in their α-helix or β-sheet binding interface. We found that the coexistence densities in the dilute phase (referred to as the saturation concentration, $C_{sat}$) of the two configurations of HP1α homodimer were comparable within the error of three independent simulations (*Figure 2—figure supplement 1k*). This suggests that the dimerization mode of the CSD domain does not affect the phase separation

capability of the HP1 proteins as long as the homodimer can be formed. We note, however, that under physiological conditions, the phase separation propensity of the two configurations may be influenced by the variance in their binding affinities. Therefore, for the rest of the study, we will focus on the α-helix-mediated CSD-CSD conformation as the primary and biologically relevant form of HP1 homodimerization.

Under the simulation conditions, HP1β did not undergo phase separation, while HP1α and HP1γ formed stable condensates (as evidenced in the flat density profiles in *Figure 2f* and snapshots in *Figure 2g*; *Figure 2—video 2*), but the $C_{sat}$ of HP1γ was approximately twofold higher than that of HP1α (inset of *Figure 2f*). We also constructed the phase diagrams for HP1 paralogs and found that the critical temperature (or the phase separation propensity) decreased in order for HP1α, HP1γ, and HP1β (*Figure 2h*). This correlates with the conformational expansion trends observed in the CG homodimer simulations under dilute conditions. These observations are consistent with experimental data and the relationship between single-molecule dimensions and phase separation propensity (*Larson et al., 2017*; *Qin et al., 2021*; *Dignon et al., 2018a*). Notably, in the absence of phosphorylation and DNA, HP1α has been reported to phase-separate at a much lower protein concentration than HP1γ in vitro, while HP1β did not form phase-separated droplets under any tested experimental conditions. Moreover, SAXS data have shown that HP1α conformations are more compact than HP1β structures (*Larson et al., 2017*; *Munari et al., 2013*). To date, the SAXS data for HP1γ has not been reported. Under crowding conditions, the conformational differences among HP1 paralogs are less pronounced (*Figure 2i*).

To explore the interactions that mediate phase separation of HP1 paralogs, we computed 1D and 2D intermolecular (between homodimers) contact maps within the condensed phase. We found that the number of intermolecular contacts at the domain level of HP1β homodimers was much lower compared to HP1α and HP1γ, as expected (*Figure 2l*). Moreover, HP1β does not show noticeable intermolecular interactions between domains that are crucial to establishing multivalent contacts (*Figure 2l*). On the contrary, NTE-hinge and CTE-hinge interactions are dramatically highlighted in the 2D contact map of HP1α (*Figure 2k*). These interaction patterns are also present in HP1γ, but the overall number of contacts per domain is less than the number for HP1α (*Figure 2m and j*). Notably, the inter-dimer NTE1-NTE2 contacts of HP1γ are substantially enhanced due to the presence of the extended charge-segregated diblock in its NTE (*Figures 2m and 1i*).

In summary, NTE-hinge interactions and NTE-NTE (between the charge-segregated diblocks) interactions govern the conformation of HP1 paralogs in the dilute state and drive multivalency under crowding conditions. Additionally, substantial interactions between disordered segments (NTE, hinge, and CTE) and folded domains (CD and CSD) highlight the positive role of folded domains in the condensate formation of HP1 paralogs (*Figure 3a*).

## Disordered and folded domains cooperatively mediate phase separation of HP1 proteins

Multi-domain proteins have been shown to exhibit complex multivalent interaction networks that mediate phase separation (*Mohanty et al., 2022*). Previous research has demonstrated the influence of electrostatics on the interactions between IDRs and folded domains (*Martin et al., 2021*). For example, it has been observed that folded domains affect the behavior of IDRs in a manner dependent on the accessibility of charged residues on the surface of the folded domain (*Taneja and Holehouse, 2021*) and the charge properties of IDR sequences (*Dignon et al., 2018a*). In studies of HP1α phase separation, the less conserved disordered regions (hinge, NTE, and CTE) are thought to be sufficient to establish multivalent contacts for higher-order oligomerization, but the role of the highly conserved folded domains (CD and CSD) has not been explored (*Tibble and Gross, 2023*). Given that the CD and CSD contain a significant number of charged residues and possess high net negative charge, we asked whether electrostatic interactions influence the interplay between IDRs and folded domains in the phase separation of HP1 paralogs. To explore this, we performed domain-swapping phase coexistence simulations for the three HP1 paralogs.

Given that HP1α has the highest phase separation propensity, we first performed swaps of the IDRs (NTE, hinge, and CTE) of HP1β and HP1γ into HP1α, yielding constructs HP1α-βIDRs and HP1α-γIDRs (*Supplementary file 1*), respectively. We found that the HP1β or HP1γ IDRs fail to enhance inter-dimer NTE-hinge interactions. These chimeras are less prone to phase separation compared to the wild-type

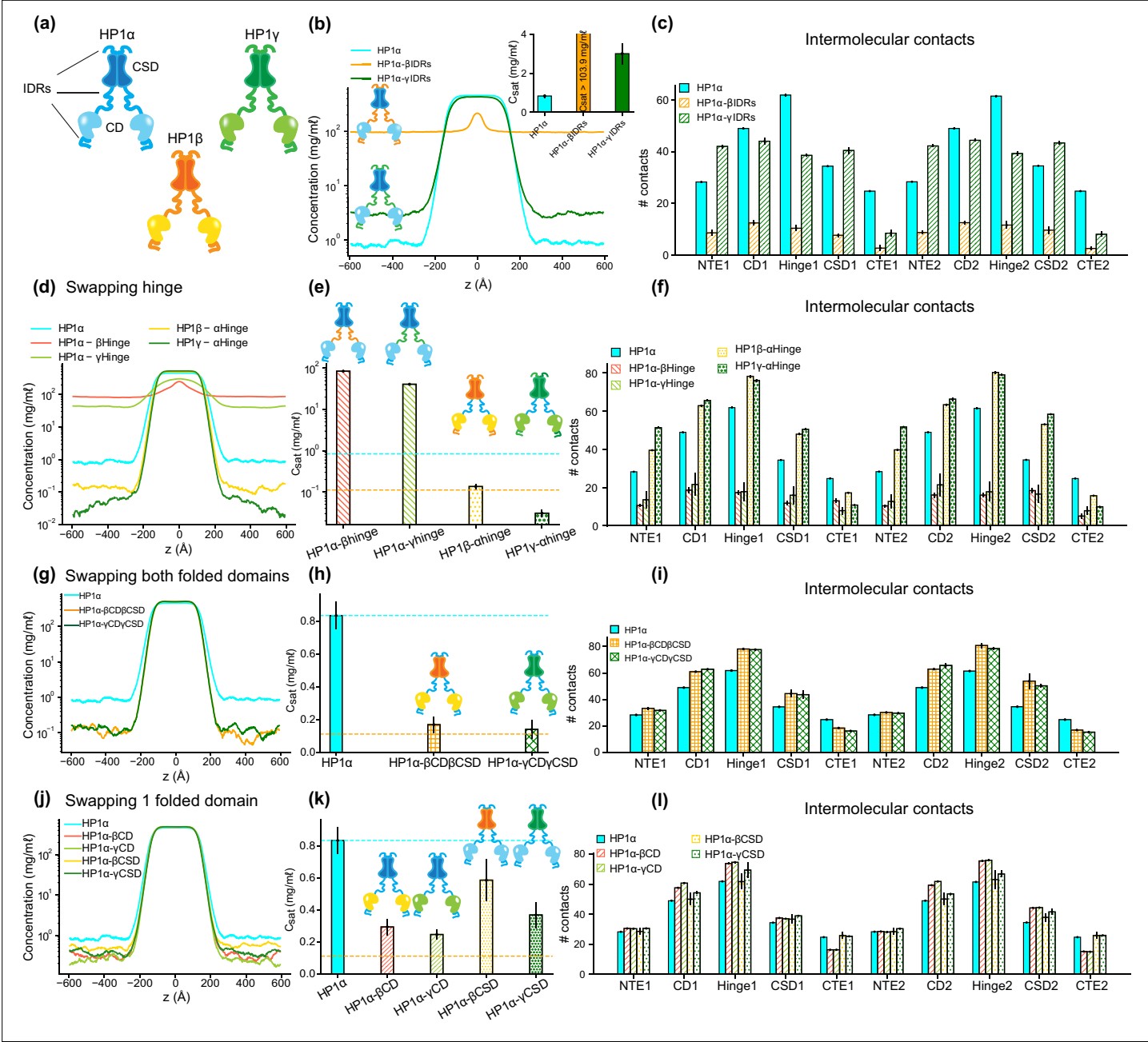

**Figure 3.** The interplay of intrinsically disordered regions (IDRs) and folded domains in mediating phase separation of heterochromatin protein 1 (HP1) paralogs. (a) Cartoons representing the three human HP1 paralogs. (b, c) Density profiles, saturation concentrations (inset), and average contacts made by each domain of HP1α with IDRs replaced with those from HP1β or HP1γ. (d, e, f) Density profiles, saturation concentrations, and average contacts made by each domain of HP1α chimeras whose hinge was swapped with the hinge from either HP1β or HP1γ(HP1α-βHinge and HP1α-γHinge, respectively),and HP1β and HP1γ chimeras whose hinge regions were replaced with the hinge of HP1α (HP1β-αHinge or HP1γ-αHinge, respectively). (g, h, i) Density profiles, saturation concentrations, and average contacts made by each domain of HP1α chimeras with folded domains replaced with those from HP1β or HP1γ (HP1α-βCDβCSD and HP1α-γCDγCSD, respectively). (j, k, l) Density profiles, saturation concentrations, and average contacts made by each domain of HP1α chimeras with either chromodomain (CD) or chromoshadow domain (CSD) replaced with the corresponding domain from HP1β (HP1α-βCD and HP1α-βCSD, respectively) or HP1γ (HP1α-γCD and HP1α-γCSD, respectively). The cyan and orange dashed lines show the simulated saturation concentrations of wild-type HP1α and pHP1α, respectively. The error bars represent the standard deviation from triplicate simulation sets. The coarse-grain (CG) coexistence simulations were conducted using the HPS-Urry model at 320 K and 100 mM salt concentration.

The online version of this article includes the following figure supplement(s) for figure 3:

**Figure supplement 1.** Intermolecular interactions of heterochromatin protein 1 (HP1) chimeras within the condensed phase.

HP1α (*Figure 3b and c*). In fact, HP1α-βIDRs did not undergo phase separation, and the $C_{sat}$ of HP1α-γIDRs is twofold greater than the concentration for HP1α. The results indicate that the IDRs of HP1α are crucial to establishing multivalent contacts for condensate formation.

We next performed hinge-swapping simulations to explore how individual hinge regions regulate phase separation in the absence of DNA. We proceeded with replacing the hinge of HP1α with the corresponding hinge segment from either HP1β or HP1γ (constructs HP1α-βHinge and HP1α-γHinge in *Supplementary file 1*, respectively). We found that neither of the chimeras formed stable condensates (*Figure 3d and e*). The hinges of HP1β and HP1γ failed to provide the necessary multivalent contacts with other HP1α domains (*Figure 3f*) due to insufficient segregation of basic blocks compared to the charge pattern of the HP1α hinge (*Figure 1g–i*). Indeed, engineered HP1β, where four acidic amino acids in the hinge region were replaced with basic residues to provide an additional stretch of positive charge, underwent phase separation at protein concentrations comparable to HP1α (*Qin et al., 2021*). Taken together, these results demonstrate that the positively charged lysine/arginine residue clusters in the HP1α hinge are necessary to induce inter-dimer contacts for condensate formation.

We next performed swaps of the HP1α hinge into HP1β (construct HP1β-αHinge, *Supplementary file 1*) and HP1γ (construct HP1γ-αHinge, *Supplementary file 1a*). We found that both chimeras readily underwent phase separation (*Figure 3d*). The HP1β-αHinge chimera displayed $C_{sat}$ that was lower than that for wild-type HP1α, and that was comparable to the $C_{sat}$ of pHP1α, which is much more prone to phase separation (*Her et al., 2022*). HP1γ-αHinge showed an even more remarkable reduction in the $C_{sat}$, which was nearly threefold lower compared to pHP1α (*Figure 3e*). The presence of the HP1α hinge led to increased multivalency in almost all domains of the chimeras (*Figure 3f*), particularly magnifying NTE-hinge interactions (*Figure 3—figure supplement 1b and c*). The main exception was the CTE region, where contacts were reduced on average. The charge segregation in the HP1β CTE and the relatively short CTE in HP1γ may account for the reduced number of contacts in this part of the sequence. These results align well with HP1α hinge-swapping experiments between different HP1 paralogs in the presence of DNA, which suggests that the HP1α hinge can make more contact with the NTE when there is reduced interference from the CTE (*Keenen et al., 2021*). We also observed a substantial increase in the inter-dimer contacts in the CD and the CSD (*Figure 3f*), raising the question of whether the folded domains may also influence the propensity for phase separation outside their wild-type context.

To investigate this question, we proceeded by inserting the folded domains of either HP1β or HP1γ into HP1α (constructs HP1α-βCDβCSD and HP1α-γCDγCSD in *Supplementary file 1a*, respectively). Intriguingly, we again observed that both chimeras readily formed stable condensates, and their $C_{sat}$ values were lower than the concentration for HP1α and comparable to the concentration for pHP1α (*Figure 3g and h*). Multivalent contacts considerably increased for the folded domains (CD and CSD) and the hinge, but markedly decreased for the CTE region (*Figure 3i*), although overall these chimeras share similar interaction patterns as HP1α (*Figure 3—figure supplement 1d and e*). These results suggest that folded domains from either HP1β or HP1γ produce favorable interactions between the HP1α hinge and the swapped folded domains while disturbing the hinge-CTE interaction.

We, therefore, wondered if these effects stemmed from either the CD or the CSD alone, or collectively from both folded domains. To test these possibilities, we separately replaced the CD and CSD of HP1α with the corresponding folded domain from either HP1β (constructs HP1α-βCD and HP1α-βCSD, respectively) or HP1γ (constructs HP1α-γCD and HP1α-γCSD in *Supplementary file 1a*, respectively). We found that swapping the CD domain alone (from either HP1β or HP1γ) substantially enhanced interactions between the hinge and the folded domains and reduced competing CTE contacts, hence promoting phase separation more efficiently. Despite sharing high sequence identity and domain architecture similarity, the CD domains of HP1β and HP1γ contain more negatively charged residues with a larger solvent-accessible surface area (SASA) than HP1α (*Figure 3—figure supplement 1f and g*). The CD and the NTE may cooperatively facilitate multivalent interactions with the hinge and substantially contribute to the overall interactions that stabilize the protein condensed phase (*Figure 3l*; *Figure 3—figure supplement 1h and i*). On the other hand, swapping of the CSD slightly enhanced phase separation compared to wild-type HP1α. The interactions in the folded domains and the hinge were slightly increased, but the interactions from the CTE remained similar to those of wild-type HP1α (*Figure 3l*; *Figure 3—figure supplement 1j,k*). Therefore, our data suggest that the CD

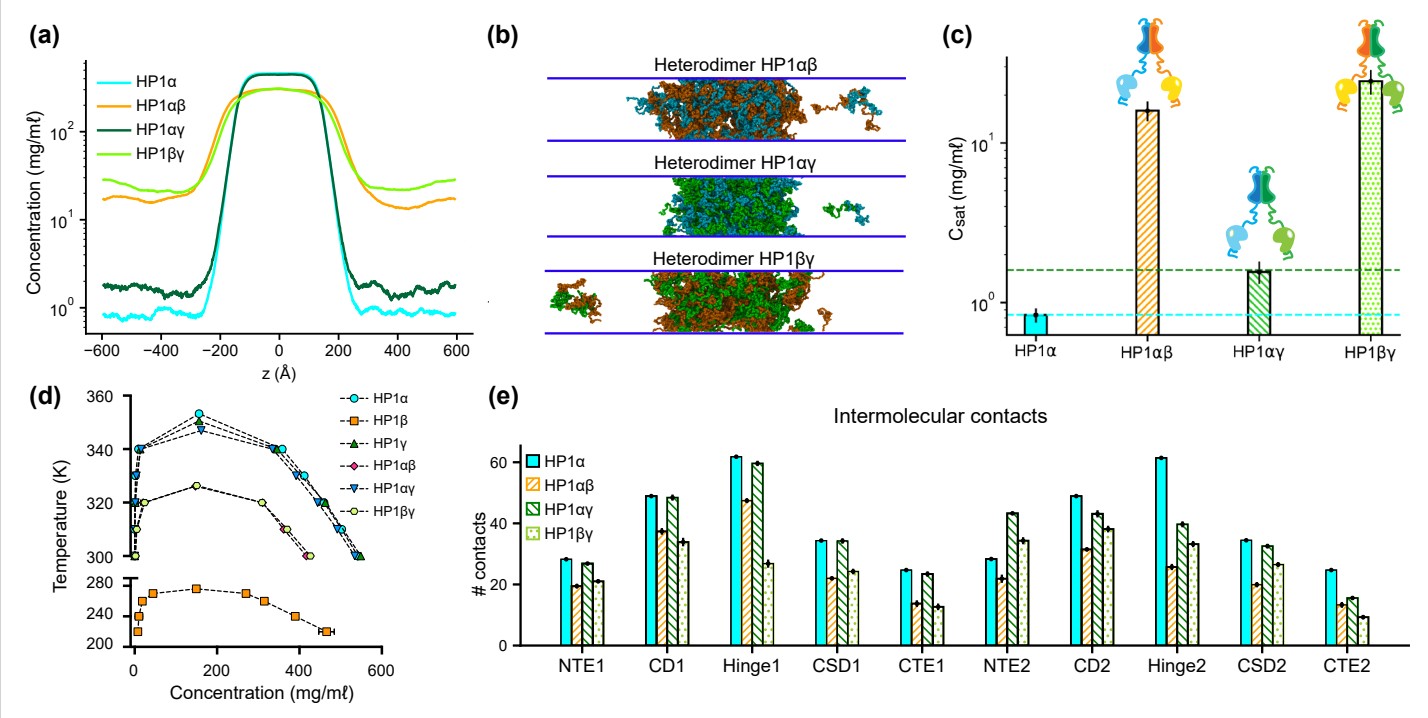

**Figure 4.** Phase separation of heterochromatin protein 1 (HP1) heterodimers. (**a, b, c**) Density profiles, snapshots of the condensates, and saturation concentrations in the coarse-grain (CG) coexistence simulations performed with heterodimers HP1αβ, HP1αγ, and HP1βγ. (**d**) Phase diagram of HP1 heterodimer phase separation conducted at different temperatures. (**e**) Average intermolecular contacts for each domain of the HP1 heterodimers in CG coexistence simulations. The error bars represent the standard deviation from triplicate simulation sets. The CG coexistence simulations were conducted using the HPS-Urry model at 320 K and 100 mM salt concentration.

The online version of this article includes the following figure supplement(s) for figure 4:

**Figure supplement 1.** Intermolecular interactions of heterochromatin protein 1 (HP1) heterodimers within the condensed phase.

domain is more dominant than the CSD in influencing the inter-domain interactions that drive phase separation.

Taken together, our findings reveal the cooperative roles of disordered and folded domains in mediating the condensation behavior of HP1 paralogs. While the condensate formation of HP1 mainly relies on NTE-hinge interactions, the folded domains, particularly the CD domain, also substantially contribute to the multivalent interactions required for the formation of oligomeric networks and phase separation. Moreover, the observations that different domains can influence multivalency across paralog chimeras made us wonder whether we may observe similar effects in HP1 paralog heterodimers.

## The HP1α monomer facilitates multivalent interactions in heterodimer phase separation

Previous literature has shown that the CSD-CSD dimerization of HP1α occurs with an estimated $K_d$ in the nanomolar range (*Her et al., 2022*; *Brasher et al., 2000*). Moreover, the CSDs of HP1 paralogs exhibit a high degree of homology, suggesting that heterodimerization is likely to occur. Indeed, in vivo and in vitro co-immunoprecipitation studies have indicated that mammalian HP1 paralogs can directly interact with each other to form heterodimers in solution and on chromatin (*Nielsen et al., 2001*).

To explore the phase separation propensity of the heterodimers, we first used MODELLER to perform homology modeling with the HP1α structure as the template to assemble heterodimers between HP1α and HP1β or HP1γ (HP1αβ and HP1αγ, respectively), and heterodimers between HP1β and HP1γ (HP1βγ). We next conducted coexistence phase simulations and calculated the coexistence densities and the intermolecular contact maps of the three heterodimers. We found that the

three heterodimers formed stable condensates (*Figure 4a and b*). The $C_{sat}$ for HP1αβ and HP1αγ heterodimers were nearly tenfold higher than the concentration for HP1α homodimers, while the $C_{sat}$ of HP1αγ heterodimers was comparable to the concentration of HP1γ homodimers (*Figure 4a–c*). We also constructed the phase diagrams for the three heterodimers and plotted them together with the homodimers. The critical temperature for HP1αγ heterodimers (347.0 K) is slightly lower than the temperatures for HP1γ homodimers (350.5 K) and HP1α homodimers (353.3 K). The critical temperatures of HP1αβ and HP1βγ heterodimers (326.1 K and 326.4 K, respectively) are substantially lower but not as low as the temperature for HP1β homodimers (276.1 K). It appears that the presence of either HP1α or HP1γ monomer in heterodimerization with HP1β can induce the necessary inter-dimer interactions in HP1β monomers to sufficiently stabilize the condensed phase (*Figure 4e*; *Figure 4—figure supplement 1a and b*).

On the other hand, the presence of HP1β monomer globally decreases inter-dimer contacts for the other partner, HP1α or HP1γ, compared to their respective homodimer contacts (*Figure 4—figure supplement 1a and b*). We also observed that the presence of the HP1γ monomer in the HP1αγ heterodimers did not alter the inter-dimer interaction patterns of the HP1α monomer, while the interactions in the HP1γ hinge became less favorable than the interactions in its respective homodimer (*Figure 4e*; *Figure 4—figure supplement 1c*). This could be due to competition with the HP1α hinge in establishing multivalent contacts. Overall, the results demonstrate the phase separation competence of HP1 heterodimers, raising the possibility that heterodimerization plays an essential role in regulating the co-localization and/or the function of HP1 paralogs. We also note that due to the treatment of folded domains as rigid bodies and the spatial constraints imposed on the CSD dimer, our simulations may not fully capture the biological behavior of HP1 heterodimers.

## DNA mediates phase separation of HP1α homo- and heterodimers

Recent studies have demonstrated that HP1α can interact with DNA via stretches of basic residues located within the hinge region, mediating the bridging of distinct DNA regions and inducing DNA compaction (*Keenen et al., 2021*; *Larson et al., 2017*). Furthermore, HP1α-DNA interactions may induce a local increase in HP1α concentration, thereby facilitating the formation of higher-order HP1α oligomers and resulting in condensate formation. HP1γ has also been shown to form condensates with DNA but requires higher protein concentrations to induce droplet formation than HP1α (*Keenen et al., 2021*). Conversely, HP1β did not undergo condensation with DNA under any tested conditions (*Keenen et al., 2021*). Inspired by these findings, we further explored the molecular interactions that dictate the phase behavior differences among HP1 homodimers and tested the phase separation competence of HP1 heterodimers with DNA.

To explore HP1-DNA interactions, we initiated condensate formation with the homo- and heterodimer systems described above and added a small mole fraction of dsDNA. Using a recently developed nucleic acid model, we first conducted CG coexistence phase simulations of each HP1 homodimer containing 0.01 mole fraction of 147 bp dsDNA. This system is equivalent to an experimental system containing 50 μM HP1 protein with 0.6 μM of 147 bp dsDNA (*Supplementary file 1b*). We found that the HP1β homodimer did not undergo phase separation, while dsDNA partitioned into and stabilized the condensates of HP1α and HP1γ (*Figure 5a and b*; *Figure 5—video 1*). In the presence of dsDNA, the $C_{sat}$ of HP1γ is approximately twofold higher than that of HP1α. It should be noted that increasing the mole fraction of dsDNA in the mixture may significantly enhance the phase separation of HP1α. Still, excessive addition of dsDNA may lead to unstable condensate due to a large increase of the overall net negative charge as observed in our previous simulations (*Her et al., 2022*). To characterize the protein-DNA molecular interactions, we computed the intermolecular contact maps based on vdW contacts formed between HP1 homodimers and dsDNA as a function of residue number. We found that the electrostatic interactions between dsDNA and the lysine/arginine-rich hinge dominated in the HP1α homodimer (*Figure 5c*). dsDNA also favored the long stretch of lysine-rich NTE of HP1γ homodimers (*Figure 5e*). However, the electrostatic interactions between dsDNA and HP1β were unfavorable due to extended patches of acidic residues along the HP1β sequence (*Figure 5d*).

We next investigated the phase separation of HP1 heterodimers with dsDNA using the simulation protocol above. We observed that DNA could induce condensate formation of all HP1 heterodimers (*Figure 5f and g*). The $C_{sat}$ values for HP1αγ, HP1αβ, and HP1βγ heterodimers in the presence of dsDNA increased in progressive order and followed a similar trend as the heterodimers alone (insets

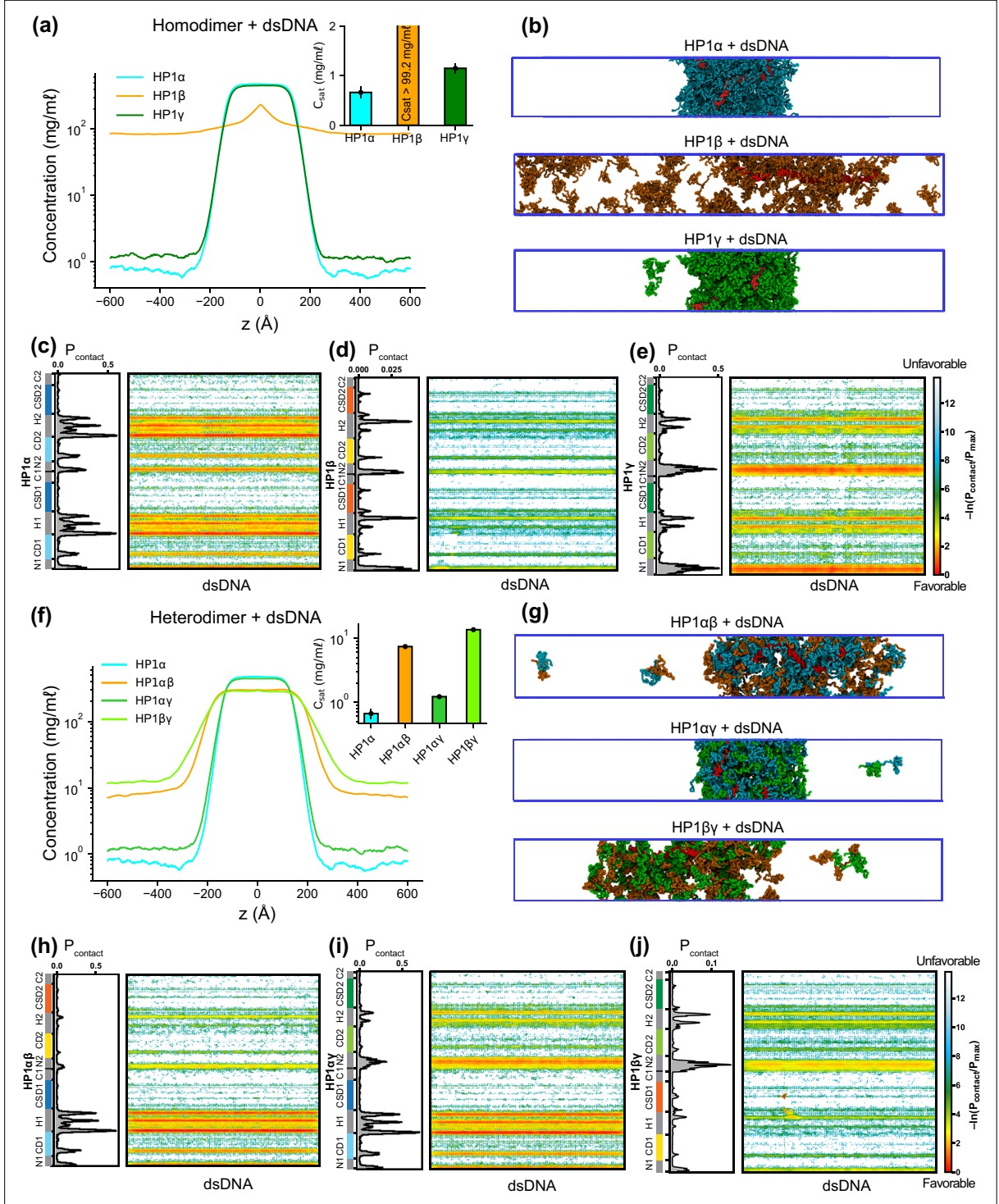

**Figure 5.** Phase separation of heterochromatin protein 1 (HP1) homo- and heterodimers with DNA. (**a, b**) Density profiles, saturation concentrations (inset), and snapshots of HP1 paralogs with one chain of 147 bp double-stranded DNA in the coarse-grain (CG) coexistence simulations. (**c, d, e**) Intermolecular contacts between HP1 homodimers and DNA. (**f, g**) Density profiles, saturation concentrations (inset), and snapshots of HP1 heterodimers with one chain of 147 bp double-stranded DNA in the CG coexistence simulations. (**h, i, j**) Intermolecular contacts between HP1 heterodimers and DNA. Preferential interactions between HP1 protein and DNA are shown in red. The error bars represent the standard deviation from triplicate simulation sets. The CG coexistence simulations were conducted using the HPS-Urry model at 320 K and 100 mM salt concentration.

The online version of this article includes the following video for figure 5:

**Figure 5—video 1.** Coarse-grain (CG) phase coexistence simulations of heterochromatin protein 1 (HP1) paralogs with dsDNA.
https://elifesciences.org/articles/90820/figures#fig5video1

Biochemistry and Chemical Biology | Structural Biology and Molecular Biophysics

of *Figure 5f and a*). The presence of HP1α monomer in complex with either HP1β or HP1γ helped facilitate the hinge-DNA interactions necessary for condensate formation. However, the considerable net negative charge of the HP1β monomer weakened the multivalent interaction network, resulting in expansion and instability within the condensates of HP1αβ and HP1βγ heterodimers (*Figure 5g, h and j*). The NTE of the HP1γ monomer showed strong binding to DNA and became the main contributor to the condensate formation of HP1βγ heterodimers with DNA. However, in HP1αγ heterodimers, this interaction competed with the favorable HP1α hinge-DNA interactions (*Figure 5i*). Overall, the results highlight the emergence of favorable electrostatic attraction between DNA and patches of basic residues in the HP1α hinge and the HP1γ NTE and unfavorable electrostatic repulsion between DNA and HP1β domains within HP1 homo- and heterodimer condensates. These observations have implications for understanding how DNA mediates phase separation of HP1 proteins in the context of heterochromatin organization.

## DNA influences the co-localization of HP1 paralogs

Although HP1 paralogs have been found in different regions of the genome (*Minc et al., 1999*; *Lomberk et al., 2006b*; *Eberhart et al., 2013*; *Schoelz and Riddle, 2022*), they localize predominantly to heterochromatic domains and play a vital role in the formation and stabilization of higher-order chromatin structures, which modulate gene expression. Moreover, HP1α has been shown to form phase-separated droplets and bind most strongly to DNA, followed by HP1γ and then HP1β (*Keenen et al., 2021*; *Larson et al., 2017*). Given that HP1 paralogs show different phase separation behaviors, we wondered how HP1 paralogs behave when they are co-localized in overlapping genomic regions. To explore the co-localization capability of HP1 paralogs, we performed CG coexistence phase simulations of mixtures of HP1α and HP1β homodimers and HP1α and HP1γ homodimers, respectively. Depending on the cell type, HP1β is considered to be less abundant than HP1α, while HP1γ may be present at comparable levels to HP1α (*Bártová et al., 2005*). Therefore, in our simulations, we started with equimolar ratios of the mixed populations (*Figure 6—video 1*) and then decreased the concentration of either HP1β or HP1γ while maintaining the same total concentration of proteins (*Figure 6a*). We next built the phase diagrams of this two-component system to explore how HP1β and HP1γ partition into HP1α condensates.

We found that the HP1β/HP1α homodimer mixtures showed behavior consistent with scaffold-client co-phase separation (*Figure 6b*). HP1α functioned as a scaffold molecule that phase separated on its own, and HP1β acted as a client that was not able to undergo phase separation on its own. Still, it had favorable heterotypic interactions with HP1α and was incorporated into HP1α condensates. Slowly increasing the HP1β concentration in the system did not significantly affect the phase separation propensity of HP1α (its dilute-phase concentration remained relatively the same ~1 mg/ml), and HP1β was recruited to the HP1α condensate. However, the excessive presence of HP1β led to competition between the self-interaction of HP1α-HP1α and the cross-interaction of HP1β-HP1α, resulting in increased dilute-phase concentration for both HP1α and HP1β (*Figure 6b*, trend in lower left corner of the graph). In contrast, the HP1γ/HP1α homodimer assembly showed behavior consistent with cooperative co-phase separation (*Figure 6c*). Both HP1α and HP1γ have a strong homotypic affinity, and they were able to phase separate on their own and cooperatively form heterotypic condensates. This behavior is attributed to the similarity of the two paralogs in sequence properties and phase separation behaviors. However, the condensate was more enriched in the HP1α component as it displays a stronger self-interaction propensity compared to HP1γ. In the case of equimolar initial concentration, the dilute-phase concentration of HP1γ was nearly twofold higher than that of HP1α (*Figure 6c*, green circle in lower left corner of the graph).

Given that HP1 paralogs show different preferences in DNA binding, we next asked whether the presence of DNA may regulate the co-localization of HP1 paralogs. We followed the simulation protocol described above and performed simulations in the presence of 0.01 mole fraction of 147 bp dsDNA. These simulations were initiated from a high-density slab, where the two HP1 paralogs and dsDNA were mixed. We first investigated the stability of the condensates of equimolar mixtures in the presence of dsDNA. We observed that HP1β homodimers were predominantly localized towards the periphery of the condensate while HP1α and dsDNA occupied the interior (*Figure 6d and e*; *Figure 6—video 2*). On the other hand, HP1γ was able to co-localize and form stable condensates with HP1α and dsDNA (*Figure 6d and f*; *Figure 6—video 2*). We next constructed the two-component

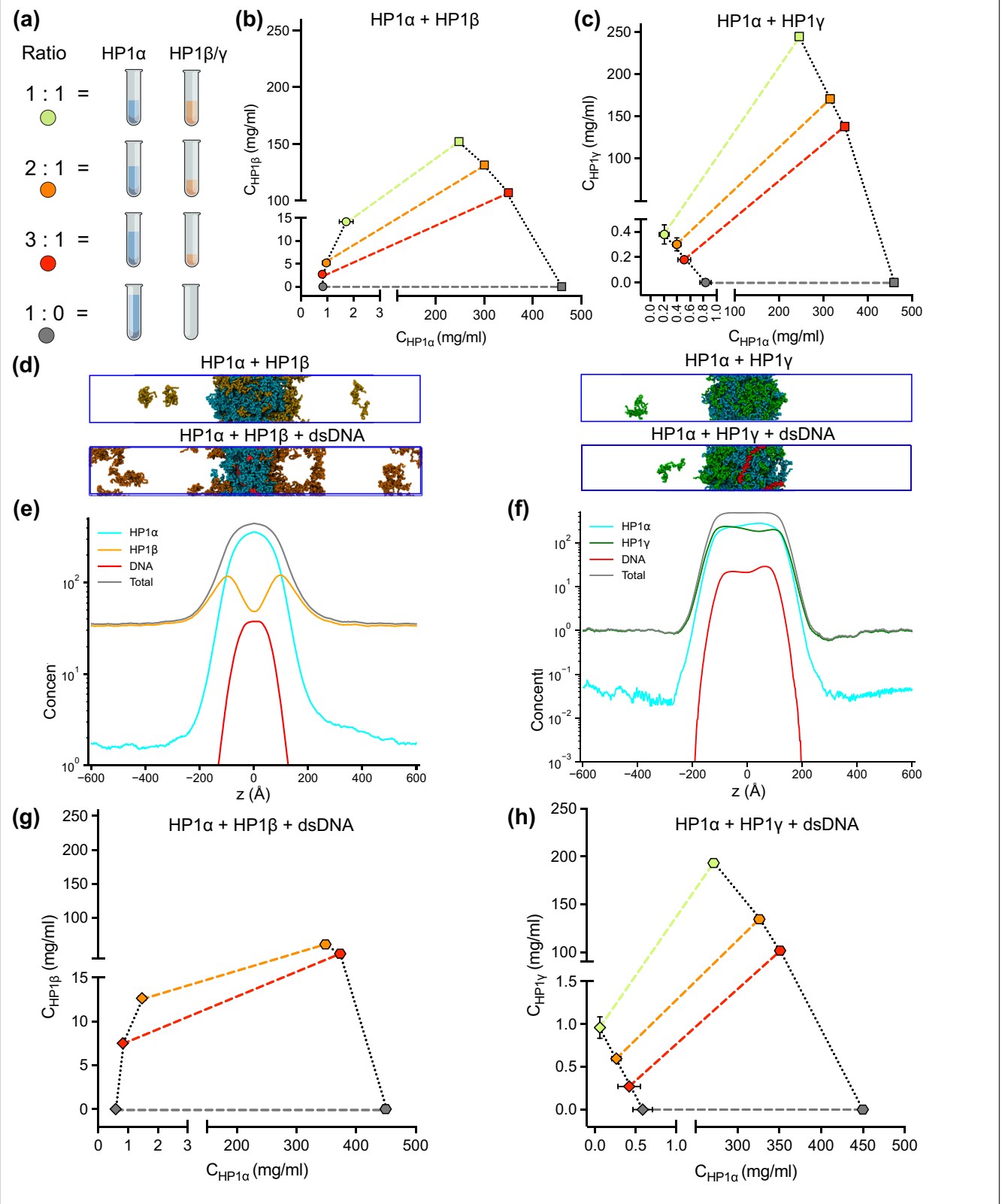

**Figure 6.** DNA regulates the condensation of heterochromatin protein 1 (HP1) paralog mixtures. (**a**) Schematic of the ratios of HP1α and HP1β (or HP1γ) used to generate multicomponent phase diagrams. The total concentration of the mixture was fixed in all cases. (**b, c**) Multicomponent phase diagrams of HP1α mixing with HP1β (or HP1γ). The color code corresponds to the ratios shown in the schematic in (**a**). The circles and the squares show the protein concentrations in the dilute and the dense phases, respectively. The tie line (colored dashed line) connects the concentrations in the dense

*Figure 6 continued on next page*

*Figure 6 continued*

and dilute phases for each ratio. Phase existence lines (black dotted lines) define the two arms of the phase diagram at low and high concentrations. (**d**) Snapshots of the condensates in the coarse-grain (CG) coexistence simulations of HP1α+HP1β and HP1α+HP1γ mixtures at an equimolar concentration in the absence and presence of dsDNA. (**e, f**) The density profiles of the mixtures at an equimolar concentration in the presence of dsDNA. (**g, h**) Multicomponent phase diagrams of HP1α mixing with HP1β or HP1γ in the presence of DNA. The 1:1 ratio data points for HP1α and HP1β mixing with DNA were excluded in (**g**) due to the instability of the condensate. The diamonds and the hexagons show the protein concentrations in the dilute and the dense phases, respectively. The error bars represent the standard deviation from triplicate simulation sets. The CG coexistence simulations were conducted using the HPS-Urry model at 320 K and 100 mM salt concentration.

The online version of this article includes the following video(s) for figure 6:

**Figure 6—video 1.** Coarse-grain (CG) phase coexistence simulations of HP1α+HP1β and HP1α+HP1γ mixtures at an equimolar concentration.
https://elifesciences.org/articles/90820/figures#fig6video1

**Figure 6—video 2.** Coarse-grain (CG) phase coexistence simulations of HP1α+HP1β and HP1α+HP1γ mixtures at an equimolar concentration in the presence of dsDNA.
https://elifesciences.org/articles/90820/figures#fig6video2

phase diagrams of HP1α+HP1β and HP1α+HP1γ mixtures in the presence of dsDNA (*Figure 6g,h*). We found that HP1α phase separation was slightly enhanced in the presence of DNA, while HP1β was more excluded from the HP1α-dsDNA core due to its electrostatic repulsion with dsDNA. By contrast, dsDNA did not affect the overall cooperative phase separation behavior of HP1α and HP1γ. However, compared to the case of mixing HP1α and HP1γ alone, the dilute-phase concentration of HP1γ was found to be twofold higher, whereas the dilute-phase concentration of HP1α was nearly twofold lower. In other words, the presence of dsDNA substantially enhanced the phase separation of HP1α but slightly excluded HP1γ from the HP1α-DNA complex. Together, these results suggest that HP1γ may co-partition in HP1α-DNA condensates while HP1β mostly remains in the condensate boundary. The observations highlight the role of DNA in modulating the co-localization of HP1 paralogs, which carries significant implications for understanding their distinct biological functions in the context of chromatin.

## Discussion

HP1 paralogs are an essential modulator of chromatin architecture, gene regulation, and genomic integrity in mammalian cells (*Lachner et al., 2001*; *Bannister et al., 2001*; *Maison and Almouzni, 2004*; *Ayyanathan et al., 2003*). Despite their sequence conservation, the three human paralogs, HP1α, HP1β, and HP1γ, exhibit different nuclear localization and interaction patterns that may underlie potentially unique functions in gene silencing and heterochromatin establishment (*Kwon and Workman, 2011*; *Cowieson et al., 2000*; *Grewal and Jia, 2007*). These distinct behaviors may be intimately connected to the proteins' ability to undergo LLPS, a process that can substantially remodel the material properties of chromatin environments (*Gibson et al., 2019*; *Farr et al., 2021*). Our in-depth in silico investigation suggests that the distinct LLPS propensities of the HP1 paralogs result from the precise tuning of the charge distribution along the protein sequence.

In the absence of DNA, the main driver for LLPS is the relative strength of hinge-NTE interactions. HP1α, the most LLPS-prone paralog, has the most positively charged hinge region, which can interact with negatively charged patches on the NTE and the CTE. While hinge-NTE interactions are dominant, competition from the CTE negatively influences the LLPS propensity of the wild-type protein. This balance can be shifted dramatically by the phosphorylation of serine residues 11–14 on the NTE, which significantly enhances LLPS (*Her et al., 2022*; *Larson et al., 2017*). In HP1γ, on the other hand, the NTE contains a relatively long stretch of positively charged residues that disfavors hinge-NTE interactions and LLPS. HP1β has the most negative overall charge of the three paralogs, with a significant number of negative residues in the hinge and a much lower propensity to form hinge-NTE interactions. Importantly, HP1γ and HP1β do not have the phosphorylatable serine patch in their NTEs and, therefore, lack the ability to tune the strength of hinge-NTE interactions. However, when the HP1α hinge was swapped into HP1β and HP1γ, it could enhance global inter-domain interactions (*Figure 3f*; *Figure 3—figure supplement 1b and c*), especially between the NTE and hinge regions. In this case, the extended stretches of negatively charged residues in the NTE of HP1β considerably enhanced NTE-hinge interactions and promoted phase separation (*Figure 3e and*

*f*; *Figure 3—figure supplement 1b*), similar to the effect of NTE phosphorylation in HP1α (*Her et al., 2022*). In both chimeras, contacts in the CTE regions were significantly reduced compared to the wild-type HP1α (*Figure 3f*), implying that reducing access of the CTE to the hinge dramatically enhances phase separation.

These trends are reflected in the $C_{sat}$ determined in our simulations and concur well with experimental findings. For example, in vitro experiments have demonstrated that, in the absence of phosphorylation and DNA, HP1α undergoes LLPS at elevated protein concentration and reduced temperature (50–400 µM, 4 °C) (*Keenen et al., 2021*; *Qin et al., 2021*) or under low salt concentration (50 mM KCl) (*Wang et al., 2019*). Our CG simulations corroborate these experimental observations, indicating that a low salt concentration (50 mM) promotes the LLPS of HP1α. Raising the salt concentration weakens the electrostatic interactions and increases the $C_{sat}$, eventually precluding HP1α's phase separation at high salt regimes (200–500 mM) (*Figure 2—figure supplement 2*). Phosphorylation allows LLPS at lower protein concentrations, higher salt conditions, and increased temperatures (~50 µM, 25 °C, and 75 mM KCl) (*Her et al., 2022*; *Larson et al., 2017*). HP1γ undergoes LLPS at a significantly higher protein concentration than HP1α (900 µM), while HP1β fails to form phase-separated droplets under any tested experimental conditions (*Qin et al., 2021*). Deletion of the HP1α CTE in the context of phosphorylation reduced the $C_{sat}$ tenfold, supporting a model where reducing competition from the CTE is key to increasing LLPS propensity. Our previous computational and experimental studies on pHP1α also agree with these conclusions (*Her et al., 2022*).

While productive and competing interactions of the IDRs of HP1 paralogs are undoubtedly the main determinants of LLPS propensities, our simulations allow us to capture the more subtle contributions of the folded CD domain. When the CD domain of HP1α was replaced with the corresponding CD domain from either HP1β or HP1γ, the phase separation of these chimeras was significantly enhanced (*Figure 3j and k*). The average interdimer contacts per region in the CD and hinge substantially increased, while contacts considerably decreased in the CTE (*Figure 3l*). Notably, the CD domains of HP1β or HP1γ contain more negatively charged residues than the HP1α CD (*Figure 1e and f*), and these residues appear to facilitate multivalent CD-hinge contacts without interfering with NTE-hinge interactions. This also indicates that the NTE and the CD can work in concert to compete with the CTE for access to the positively charged hinge. The CD of HP1 paralogs has a well-established role in recognizing and binding H3K9me2/3, facilitating the targeted recruitment and localization of HP1 to heterochromatic regions (*Wang et al., 2019*; *Schoelz and Riddle, 2022*; *Maison and Almouzni, 2004*; *Lomberk et al., 2006a*). Our simulations suggest that the CD also facilitates multivalent electrostatic interactions governing the LLPS of human HP1 paralogs. This mechanism, however, appears to be distinct from the oligomerization capabilities of Swi6 (*Sanulli et al., 2019*), which can promote multivalency through direct CD-CD interactions.

The nuclear localization patterns of HP1 paralogs (*Lachner et al., 2001*; *Bannister et al., 2001*; *Maison and Almouzni, 2004*) show that there are common overlapping regions in heterochromatin but also distinct territories where only certain paralogs are enriched, e.g., the presence of HP1γ and HP1β in euchromatin. These observations suggest the possibility that the LLPS behavior of HP1 proteins may also be influenced by the local distribution and concentration of different HP1 paralogs. The versatility of our CG model allows us to build and analyze multicomponent LLPS environments to capture this complexity and describe the dominant interactions. For example, in mixtures of HP1 paralog homodimers, the HP1α hinge region serves as a central hub, connecting adjacent paralogs through electrostatic attraction involving the NTE, CD, or CTE domains. This multivalent network enables the highly negatively charged HP1β to function as a client, partitioning into the HP1α scaffold condensate in a concentration-dependent manner. Specifically, at low concentrations, HP1β can be directly recruited to the phase-separated droplet of HP1α, consistent with experimental observations (*Larson et al., 2017*). However, increasing the presence of HP1β in the droplets activates repulsive forces between HP1β molecules, resulting in the overall instability of the condensate. In this scaffold-client phase separation context, HP1α is central to forming the condensate, with selective recruitment of HP1β dimers, which can be brought in proximity to specific genomic regions to enhance the efficiency of HP1β-mediated gene silencing or heterochromatin formation. In contrast, HP1γ can cooperatively phase-separate with HP1α due to preferential cross-interactions between its NTE/CD and the HP1α hinge region. This cooperative phase separation implies that HP1α and HP1γ co-condensation may facilitate coordinated gene regulation and chromatin organization activities.

This picture can also be complicated by the formation of heterodimers. HP1 paralogs have been shown to interact directly with each other to form heterodimers in co-immunoprecipitation experiments of mammalian cells (*Nielsen et al., 2001*). Furthermore, the high conservation of the CSD dimerization interface in the three paralogs suggests that heterodimers may form with similar affinities to homodimerization and that such species can be formed under physiological conditions. Our simulations suggest that HP1α-HP1β and HP1α-HP1γ heterodimers have a lower propensity to undergo LLPS compared to HP1α alone but display enhanced properties compared to HP1β and HP1γ homodimers, respectively. Again, the hinge region of HP1α is the key to establishing the multivalent networks in such samples. It is important to emphasize that our model is predicated on the assumption that HP1 proteins establish stable chromoshadow domain (CSD-CSD) dimers, a hypothesis supported by their $K_d$ values being in the nanomolar range (*Brasher et al., 2000*; *Ayyanathan et al., 2003*). While this simplification serves as a useful starting point, it may not fully capture the dynamic nature of HP1 dimerization. Further computational and experimental studies are needed to understand better the behavior of the complex mixtures of HP1 paralogs, particularly at phase boundaries.

Our recently developed nucleic acid model (*Kapoor et al., 2023*) also allowed us to evaluate how the interaction patterns and localization of HP1 paralogs change in the presence of DNA. In this context, the distribution of positive charge along the paralog sequence is crucial in defining LLPS. HP1α displays the most robust phase separation with DNA due to multiple attractive hotspots in the hinge, NTE, and CD (*Figure 5a and c*). HP1γ interacts with DNA primarily through its extended positively charged NTE and through some conserved basic residue stretches in the hinge (*Figure 5e*). It undergoes phase separation with DNA but is less competent than HP1α (*Figure 5a*). In HP1β, contacts with DNA are strongly disfavored due to the overall negative charge of its sequence and LLPS is not observed (*Figure 5a, b and d*). These interaction patterns provide a rationale behind the phase separation behavior observed in mixed populations of HP1 and DNA (*Keenen et al., 2021*; *Latham and Zhang, 2022*). Premixing HP1β or HP1γ with HP1α and DNA inhibits droplet formation in a concentration-dependent manner. When HP1α-DNA condensates are pre-formed, HP1β dissolves the condensates, while HP1γ stabilizes them. Our computational model suggests that when present at low protein concentrations, HP1β and HP1γ can co-localize with HP1α and partition into HP1α-DNA condensates. However, the increased presence of HP1β/HP1γ in the mixture with HP1α may activate

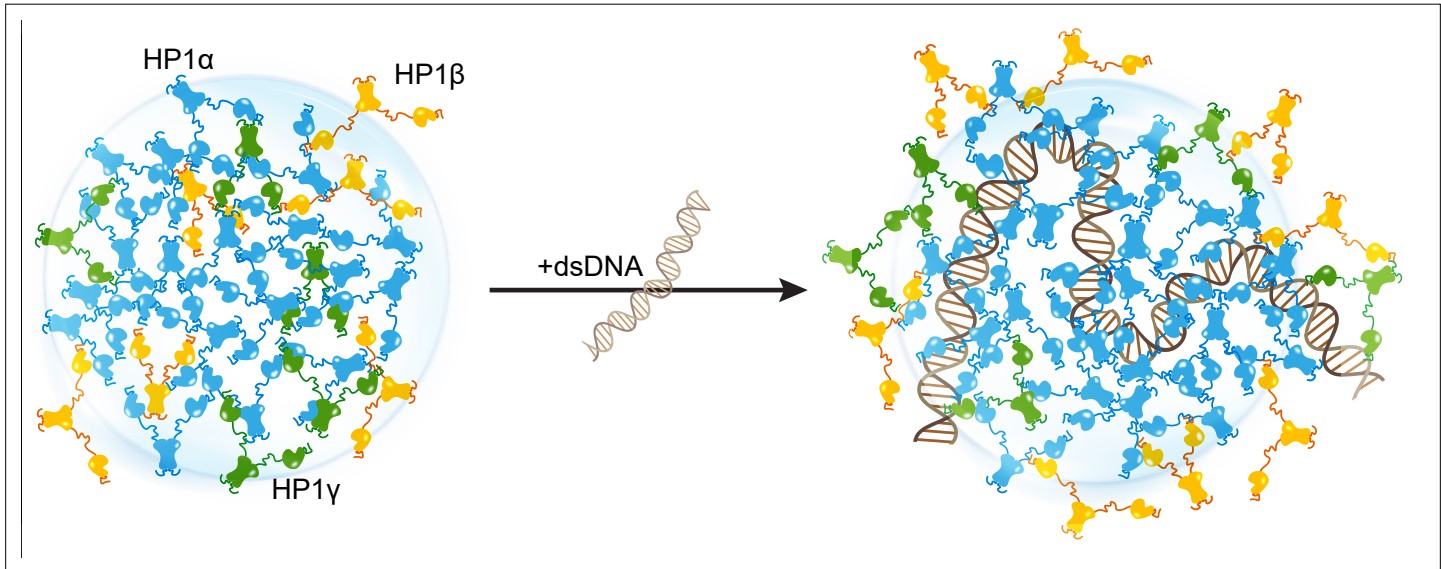

**Figure 7.** Distinct interaction patterns of heterochromatin protein 1 (HP1) paralogs influence the localization patterns of HP1α, HP1β, and HP1γ in the absence and presence of DNA.

The online version of this article includes the following video and figure supplement(s) for figure 7:

**Figure supplement 1.** Model of heterochromatin protein 1 (HP1) paralogs influences chromatin organization via liquid-liquid phase separation (LLPS).

**Figure 7—video 1.** Coarse-grain (CG) simulations of the droplet of HP1 paralogs (HP1α+HP1β+HP1γ mixture) in the absence and presence of dsDNA.
https://elifesciences.org/articles/90820/figures#fig7video1

competition for binding sites between paralog-paralog and paralog-DNA, leading to concentration-dependent phase separation inhibition. When HP1α-DNA condensates are pre-formed, and HP1β is subsequently added at equimolar concentration with HP1α, HP1β molecules mainly remain at the HP1α-DNA phase boundary. The invasion of HP1β into the droplet causes binding-site competition between HP1β-HP1α and DNA-HP1α. The repulsive forces between HP1β and DNA can further destabilize and dissolve the condensate (*Figures 5i, 6d, e and g*). In contrast, HP1γ preferentially interacts with DNA through its NTE and hinge and cooperatively phase separates with HP1α, thereby stabilizing pre-formed HP1α-DNA condensates (*Figures 5e, 6c, d and h*). It should also be noted that in HP1 paralog mixtures, heterodimerization likely occurs due to a similarly high CSD-CSD affinity (*Her et al., 2022*; *Brasher et al., 2000*), potentially altering DNA-binding propensity and regulating paralog localization with DNA.

Our in silico study paints a molecular picture describing how DNA regulates the localization patterns of HP1 proteins, which has general implications for understanding the function of HP1 proteins in heterochromatin organization and spreading. In this picture, HP1α is central in regulating DNA compaction and initiating protein condensation with DNA or upon phosphorylation. HP1γ preferentially enters and cooperatively enhances HP1α condensates, forming heterochromatic domains that silence DNA transactions. In contrast, HP1β creates a phase boundary (*Figure 7*; *Figure 7—video 1*) and fine-tunes HP1-DNA phase separation in a concentration-dependent manner, potentially limiting heterochromatin spreading. HP1β may also be recruited to more open chromatin regions to perform gene-activating associated roles in this context. Thus, our simulations suggest a model of heterochromatin organization whereby HP1α and DNA initiate condensation. HP1γ, HP1β, and heterodimerization can further enhance, dissolve, or fine-tune phase-separated droplets in a concentration-dependent fashion (*Figure 7—figure supplement 1*). This mechanism of LLPS fine-tuning is strongly dependent on the charge distribution along the HP1 paralog sequence and likely acts in concert with other components of the HP1 interaction network, including nucleosomes, H3 K9 methylation, RNA, and HP1 partners that specifically bind to the CSD-CSD dimer interface (*Canzio et al., 2014*; *Meyer-Nava et al., 2020*). Our in silico work sets the stage for dissecting the individual roles and combinatorial effects of these components in a comprehensive effort to understand the molecular forces that determine the fate of genes in the cell.

## Materials and methods
### Coarse-grained MD simulation protocol
The CG single homodimer simulations and the CG coexistence phase simulations of HP1 proteins and HP1 proteins with DNA were performed in the HOOMD-Blue 2.9.7 software package (*Anderson et al., 2020*), using the protocol described in our previous studies (*Dignon et al., 2018b*; *Regy et al., 2021b*). To simulate HP1 homodimers and heterodimers, the disordered regions (NTE, hinge, and CTE) remained flexible, whereas the folded domains were constrained using the hoomd.md.constrain.rigid function (*Nguyen et al., 2011*; *Glaser et al., 2020*). The CD domains were treated as separate rigid bodies, while the CSD-CSD domains were held as a single rigid body, allowing the molecule to mimic the dimer behavior.

The proteins and DNA were modeled using the one-bead-per-residue HPS-Urry model (*Regy et al., 2021b*) and the recently developed two-bead-per-nucleotide DNA model (*Kapoor et al., 2024*), respectively. The CG coexistence phase simulations were initiated in a slab geometry of 170 × 170 × 1190 Å$^3$ with 50 chains of dimers. In all CG simulations, a 5 µs NVT ensemble was conducted using a Langevin thermostat with a friction factor $\gamma = m_{AA}/\tau$. Here $m_{AA}$ is the mass of each amino acid bead and $\tau$ is the damping factor set to 1000 ps. The time step was set to 10 fs. As the coexistence density in the dilute phase was too low at 300 K, we simulated all systems at 320 K to compare the phase separation propensities of homo- and heterodimers and their chimeras. All the CG simulations were conducted at 100 mM salt concentration. When calculating the density profiles and contact maps, the first 1 µs of the trajectory was skipped and treated as an equilibration period.

Phase diagrams of HP1 homo- and heterodimers were constructed using a method described previously (*Dignon et al., 2018b*; *Regy et al., 2021b*). The dilute and dense phase densities ($\rho_L$ and

$\rho_H$, respectively) were extracted from the CG coexistence phase simulations. The critical temperature $T_C$ was estimated by fitting the coexistence densities at different temperatures to the following function:

$$\rho_H - \rho_L = A(T_C - T)^{\beta}$$

where $\beta$=0.325 universality class of 3D Ising model (*Rowlinson and Widom, 2013*), and A is a protein-specific fitting parameter.

Previous studies have demonstrated that slab geometry can help mitigate finite-size effects and facilitate efficient sampling of the phase diagram (*Dignon et al., 2018a*). To assess the potential impact of finite-size effects with our chosen box dimensions, we conducted a test using the HP1α homodimer, which serves as a representative system given the comparable sequence lengths of HP1 paralogs and their chimeras. By reducing the system size by 30% and constructing its phase diagram, we observed that both the original system size (50 dimers) and the reduced counterpart (35 dimers) produced similar phase diagrams, with critical temperatures of 353.3 K and 352.1 K, respectively, as shown in *Figure 2—figure supplement 3a and b*.

We further evaluated the influence of the xy cross-sectional area on the measurement of $C_{sat}$. With the z-direction box length fixed at 1190 Å³, we varied the xy cross-sectional areas (120 × 120, 150 × 150, and 200 × 200 Å²) while maintaining the protein density consistent with the control case (170 × 170 Å²). Given that HP1 dimers are multidomain proteins, a 120 × 120 Å² cross-section was the minimum size feasible to prevent particle overlap in HOOMD simulations due to the constraints of the small box size. Our findings indicate that the condensates remained stable across all tested cross-sectional areas and that there were no significant differences in $C_{sat}$ measurements within the margin of error, as depicted in *Figure 2—figure supplement 3c and d*. These results confirm that our chosen box size is sufficiently large to minimize finite-size effects, thus ensuring the robustness of our results.

In the simulations of co-phase separation between HP1 paralogs (HP1α - HP1β homodimers and HP1α - HP1γ homodimers, respectively), the total number of homodimers in the mixture was kept constant with box dimensions of 170 × 170 × 1190 Å³. The number of homodimers for each HP1 component was varied to account for different ratios of HP1α to HP1β (or HP1γ): 1, 2, 3, and HP1α only. To investigate the effects of DNA on the co-localization of HP1 paralogs, 147 bp of dsDNA was added to each system above. The multi-component phase diagrams were constructed by plotting the extracted coexisting densities of each component in the CG coexistence phase simulation at each ratio of the HP1 paralog mixture. By selecting a box size that minimizes finite-size effects, we can ensure that the spatial segregation observed in our multi-component condensates reflects genuine phase behavior. This finding aligns with (*Chew et al., 2023*), who also reported well-separated multi-layered condensates and conducted thorough validations to confirm these phases.

## All-atom MD simulation protocol

Atomistic simulations of HP1 paralogs were initially prepared using (*Páll et al., 2020*). The system topologies were modeled using the force field Amber99SBws-STQ (*Tang et al., 2020*), which is available at https://bitbucket.org/jeetain/all-atom_ff_refinements (*Phan, 2023*). Each HP1 homodimer was placed in an octahedral box of 15 nm in length. The system energy was first relaxed in a vacuum and then solvated with TIP4P/2005 water molecules (*Abascal and Vega, 2005*) using the steepest descent algorithm. To mimic the physiological salt concentration (100 mM), Na⁺ and Cl⁻ ions were added along with additional Na⁺ counter ions to achieve electrical neutrality. The improved salt parameters from Lou and Roux were used for all simulations (*Luo and Roux, 2010*). The system was first equilibrated in a canonical ensemble (NVT) using a Nose-Hoover thermostat (*Evans et al., 1985*) with a coupling constant of 1.0 ps at 300 K, then it was further equilibrated in an isothermal-isobaric ensemble (NPT) using the Berendsen barostat (*Berendsen et al., 1984*) with an isotropic coupling of 5.0 ps to achieve pressure of 1 bar. Production simulations were conducted using OpenMM 7.6 (*Eastman et al., 2017*) in the NVT ensemble at 300 K, with Langevin middle integrator (*Zhang et al., 2019*), the friction coefficient of 1 ps⁻¹, hydrogen mass increased to 1.5 amu, and 4 fs timestep. Hydrogen-related bonds were constrained using the SHAKE algorithm (*Ryckaert et al., 1977*). Short-range non-bonded interactions were calculated based on a cutoff radius of 0.9 nm, and long-range electrostatic interactions were treated using the Particle Mesh Ewald (PME) method (*Darden et al., 1998*). Distance-based contact maps were calculated using a previously published method (*Zheng et al., 2020*). Two residues

were considered to form a van der Waals contact if at least one atom of one residue was within 6 Å of an atom in the other residue.

The data supporting the findings of this study are available within the article and the *Supplementary file 1*. The 3D structures of hetero/homodimer HP1 paralogs and their chimeras, and the example scripts to perform AA and CG simulations are available on GitHub (https://github.com/TienMPhan/HP1paralogs-simulations; copy archived at *Phan, 2024*).

## Acknowledgements

We thank Dr. Utkarsh Kapoor for providing the script related to the CG 2-bead dsDNA model. We are grateful for the computational resources provided by Texas A&M High Performance Research Computing (HPRC). This work was supported by NIH grants R01GM136917 (JM) and R35GM138382 (GTD). YCK is supported by the Office of Naval Research through the U.S. Naval Research Laboratory base program.

## Additional information

### Funding

| Funder | Grant reference number | Author |
|---|---|---|
| National Institutes of Health | R01GM136917 | Jeetain Mittal |
| National Institutes of Health | R35GM138382 | Galia T Debelouchina |
| U.S. Naval Research Laboratory | | Young C Kim |

The funders had no role in study design, data collection and interpretation, or the decision to submit the work for publication.

### Author contributions

Tien M Phan, Conceptualization, Software, Formal analysis, Investigation, Visualization, Methodology, Writing – original draft, Writing – review and editing; Young C Kim, Software, Writing – original draft, Writing – review and editing; Galia T Debelouchina, Conceptualization, Supervision, Methodology, Writing – original draft, Project administration, Writing – review and editing; Jeetain Mittal, Conceptualization, Supervision, Funding acquisition, Methodology, Writing – original draft, Project administration, Writing – review and editing

### Author ORCIDs

Tien M Phan ![ORCID] http://orcid.org/0000-0002-8608-7359
Galia T Debelouchina ![ORCID] https://orcid.org/0000-0001-6775-9415
Jeetain Mittal ![ORCID] http://orcid.org/0000-0002-9725-6402

Reviewer #1 (Public review): https://doi.org/10.7554/eLife.90820.3.sa1
Reviewer #2 (Public review): https://doi.org/10.7554/eLife.90820.3.sa2
Author response https://doi.org/10.7554/eLife.90820.3.sa3

## Additional files

### Supplementary files

• Supplementary file 1. Protein and DNA sequences were used in this study.
• MDAR checklist

### Data availability

The data supporting the findings of this study are available within the article and Supplementary file 1.

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
