## [Editor Report · eLife assessment]

This **valuable** study substantially advances our understanding of molecular mechanisms driving the phase separation behavior of HP1 paralogs. The evidence supporting the conclusions is **convincing**, with rigorous and well-designed computational simulations. The work will be of broad interest to biophysicists and biochemists.

---

## [Referee Report · Reviewer #1 (Public review)]

Summary:

HP1 plays a pivotal role in orchestrating chromatin packaging through the creation of biomolecular condensates. The existence of distinct homologs offers an intriguing avenue for investigating the interplay between genetic sequence and condensate formation. In this study, the authors conducted extensive coarse-grained simulations to delve into the phase separation behavior of HP1 paralogs. Additionally, the researchers delved into the captivating possibility of various HP1 paralogs co-localizing within assemblies composed of multiple components. Importantly, the study also delved into the critical role of DNA in finely tuning this complex process.

Strengths:

I applaud the authors for their methodical approach in conducting simulations aimed at dissecting the contributions of hinges, CTE, NTE, and folded regions. The comprehensive insights unveiled in Figure 3 compellingly substantiate the significance of these protein components in facilitating the process of phase separation.

This systematic exploration has yielded several innovative revelations. Notably, the authors uncovered a nuanced interplay between the folded and disordered domains. Although disordered regions have traditionally been linked to driving phase separation through their capacity for forming multivalent interactions, the authors have demonstrated that the contribution of the CD cannot be overlooked, as it significantly impacts the saturation concentration.

The outcomes of this study serve to elucidate the intricate mechanisms and regulatory aspects governing HP1 LLPS.

---

## [Referee Report · Reviewer #2 (Public review)]

In this paper, Phan et al. investigate the properties of human HP1 paralogs, their interactions and abilities to undergo liquid-liquid phase separation. For this, they use a coarse-grained computational approach (validated with additional all-atom simulations) which allows to explore complex mixtures. Matching (wet-lab) experimental results, HP1 beta (HP1b) exhibits different properties from HP1 alpha and gamma (HP1a,g), in that it does not phase separate. Using domain switch experiments, the authors determine that the more negatively charged hinge in HP1b, compared to HP1a and HP1g, is mainly responsible for this effect. Exploring heterotypic complexes, mixtures between HP1 subtypes and DNA, the authors further show that HP1a can serve as a scaffold for HP1b to enter into condensed phases and that DNA can further stabilize phase separated compartments. Most interestingly, they show that a multicomponent mixture containing DNA, and HP1a and HP1b generates spatial separation between the HP1 paralogs: due to increased negative charge of DNA within the condensates, HP1b is pushed out and accumulates at the phase boundary. This represents an example how complex assemblies could form in the cell.

Overall, this is purely computational work, which however builds on extensive experimental results (including from the authors). The methods showcase how coarse-grained models can be employed to generate and test hypotheses how proteins can condense. Applied to HP1 proteins, the results from this tour-de-force study are consistent and convincing, within the experimental constraints. Moreover, the authors generate further models to test experimentally, in particular in light of multicomponent mixtures. Finally, the authors clearly discuss the computational methods, assumptions and limits of the methodology, which makes this a strong contribution to our understanding of biophysical basis of condensate formation in gene regulation.

---

## [Author Response]

The following is the authors’ response to the original reviews.

**Public Reviews:**

**Reviewer #1 (Public Review):**
Summary:HP1 plays a pivotal role in orchestrating chromatin packaging through the creation of biomolecular condensates. The existence of distinct homologs offers an intriguing avenue for investigating the interplay between genetic sequence and condensate formation. In this study, the authors conducted extensive coarse-grained simulations to delve into the phase separation behavior of HP1 paralogs. Additionally, the researchers delved into the captivating possibility of various HP1 paralogs co-localizing within assemblies composed of multiple components. Importantly, the study also delved into the critical role of DNA in finely tuning this complex process.Strengths:I applaud the authors for their methodical approach in conducting simulations aimed at dissecting the contributions of hinges, CTE, NTE, and folded regions. The comprehensive insights unveiled in Figure 3 compellingly substantiate the significance of these protein components in facilitating the process of phase separation.This systematic exploration has yielded several innovative revelations. Notably, the authors uncovered a nuanced interplay between the folded and disordered domains. Although disordered regions have traditionally been linked to driving phase separation through their capacity for forming multivalent interactions, the authors have demonstrated that the contribution of the CD cannot be overlooked, as it significantly impacts the saturation concentration.The outcomes of this study serve to elucidate the intricate mechanisms and regulatory aspects governing HP1 LLPS.

Weaknesses:

The authors do not provide an assessment of the quantitative precision of their model. To illustrate, HP1a is anticipated to undergo phase separation primarily under low salt concentrations. Does the model effectively capture this sensitivity to salt conditions? Regrettably, the specific salt conditions employed in the simulations are not explicitly stated. While I anticipate that numerous findings in the manuscript remain valid, it could be beneficial to acknowledge potential limitations tied to the simulations. For instance, might the absence of quantitative precision impact certain predictions, such as the CD's influence on phase separation?

We thank the reviewer for their kind feedback and for highlighting the essential mechanistic insights obtained from our study. We have addressed the concerns raised by the reviewer below, and the specific amendments made in the manuscript are also delineated.

We appreciate the reviewer's comment on our model. Our coarse-grained (CG) physics-based model integrates electrostatic and short-range interactions, parametrized based on the Urry hydrophobicity scale. This approach effectively bridges the timescale gap between simulation and experiment, offering a transferable framework to compute protein phase diagrams in temperature-concentration space that can be compared to experimental phase behavior (1). Additionally, the vdW contact probability per residue correlation between AA and CG simulations (Fig. S1 f-h) underscores our model’s capability to uncover the mechanistic insights into the phase separation of HP1 paralogs. Despite its simplicity and widespread adoption for studying sequence-dependent phase separation in biomolecular condensates, we recognize that our CG model does not yet fully replicate experimental observations or the nuanced effects of local secondary structures on phase-separation propensities. We are actively refining our methods and exploring new strategies to enhance the accuracy and efficiency of CG models for the study of biological phase separation.

In assessing the influence of salt on the LLPS of HP1α, we note that Wang et al. (2) demonstrated that HP1α can undergo LLPS at a low salt concentration (50 mM KCl). Furthermore, Wohl et al. (3) showed that the CG HPS (Kapcha-Rossky) model can capture the salt-dependent LLPS behavior through the electrostatic screening in HP1a, a *Drosophila* homolog of human HP1α. In our CG model, the salt concentration is captured by the DebyeHuckle term with tunable screening lengths, which allows for the simulations of salt-dependent effects in the low salt regime. We have added Figure S5 to illustrate the influence of salt on the LLPS propensity of HP1α. In the low-salt regime (50 mM), the Csat of HP1α was reduced by twofold compared to that at 100 mM. Increasing the salt concentration to 150 mM raised the Csat and started destabilizing the condensate. In the high salt regime (200500 mM), HP1α did not undergo phase separation, consistent with the experimental observations (2, 4–6).

**Author response image 1. sa3fig1:** Salt-dependent effects on the LLPS of HP1α homodimer. (**a, b**) Density profiles and snapshots of HP1α homodimer simulation with the box dimensions of 170x170x1190 Å3 at differing salt concentrations, 50, 100, 150, 200, 250, and 500 mM, respectively. The simulations were conducted at 320 K using the HPS-Urry model.

However, the primary objectives of our study are to elucidate the molecular interactions and to delineate the domain contributions that dictate the distinct phase-separation behaviors of the HP1 paralogs. To this end, we standardized our simulation conditions to a physiological salt concentration of 100 mM for all paralog constructs, facilitating a direct comparison and enabling physiologically relevant predictions, including those for the CD domain.We have added the salt concentration used in the CG simulations in the Materials and Methods section, relevant figure captions, and the following sentence in the third paragraph of the Discussions section to improve clarity.

“…Our CG simulations corroborate these experimental observations, indicating that a low salt concentration (50 mM) promotes the LLPS of HP1α. Raising the salt concentration weakens the electrostatic interactions and increases the Csat, eventually precluding HP1α’s phase separation at high salt regimes (200-500 mM) (Fig. S5).”

**Reviewer #2 (Public Review):**
In this paper, Phan et al. investigate the properties of human HP1 paralogs, their interactions and abilities to undergo liquid-liquid phase separation. For this, they use a coarse-grained computational approach (validated with additional all-atom simulations) which allows to explore complex mixtures. Matching (wet-lab) experimental results, HP1 beta (HP1b) exhibits different properties from HP1 alpha and gamma (HP1a,g), in that it does not phase separate. Using domain switch experiments, the authors determine that the more negatively charged hinge in HP1b, compared to HP1a and HP1g, is mainly responsible for this effect. Exploring heterotypic complexes, mixtures between HP1 subtypes and DNA, the authors further show that HP1a can serve as a scaffold for HP1b to enter into condensed phases and that DNA can further stabilize phase separated compartments. Most interestingly, they show that a multicomponent mixture containing DNA, and HP1a and HP1b generates spatial separation between the HP1 paralogs: due to increased negative charge of DNA within the condensates, HP1b is pushed out and accumulates at the phase boundary. This represents an example how complex assemblies could form in the cell.Overall, this is purely computational work, which however builds on extensive experimental results (including from the authors). The methods showcase how coarse-grained models can be employed to generate and test hypotheses how proteins can condense. Applied to HP1 proteins, the results from this tour-de-force study are consistent and convincing, within the experimental constraints. Moreover, they generate further models to test experimentally, in particular in light of multicomponent mixtures.There are, of course, some limitations to these models.First, the CG models employed probably will not be able to pick up more complex structure-driven interactions (i.e. specific binding of a peptide in a protein cleft, including defined H-bonds, or induced structural elements). Some of those interactions (i.e. beyond charge-charge or hydrophobics) may also play a role in HP1, and might be ignored here. There is also the question of specificity, i.e. how can diverse phases coexist in cells, when the only parameters are charge and hydrophobicity? Does the arrangement of charges in the NTD, hinges and CTDs matter or are only the average properties important?

We thank the reviewer for the thoughtful comments. We also appreciate the opportunity to incorporate the feedback on the reviewer’s concerns below.

We agree that the interaction picture becomes more sophisticated, and many interaction modes may be involved in the phase coexistence in the cell environment. However, due to system sizes and required sampling, studying LLPS at an atomistic resolution remains challenging with the current state-of-the-art computer hardware. Our approach employs the CG model to reduce the computational cost but still capture the predominant interactions at the residue level. We have added the plots (Fig. S1 f-h) to show the correlation of the vdW contact probability per residue for each paralog between AA and CG simulation. The Pearson correlation coefficient is approximately 0.86, suggesting a strong positive linear correlation in the contact propensity between AA and CG simulations.

**Author response image 2. sa3fig2:** 

Our sequence analysis reveals a high fraction of charged residues in HP1 paralogs, with Arg, Lys, Glu, and Asp constituting 39-45% of the total amino acid count in the sequence. This property may explain why the electrostatic interactions are predominantly involved in the phase-separation behaviors of HP1 paralogs. Our findings on electrostatically driven phase separation and co-localization of HP1 paralogs are consistent with experimental observations by Larson et al. and Keenen et al. (5, 6). Significantly, we observe that the charge patterning in the disordered regions (NTE, hinge, and CTE) plays a critical role in the LLPS of HP1 paralogs, as articulated in the second paragraph of the Discussions section. Modifying this charge patterning, such as by phosphorylating serine residues in HP1α, excising the HP1α CTE, or substituting four acidic residues with basic ones in the HP1β hinge, can profoundly augment the LLPS of these proteins (4, 5, 7). Our in silico molecular details, complemented by in vitro observations, lay a solid foundation for future experiments. These future investigations may delve deeper into the specificity of interactions and the role of structural elements in modulating HP1 phase separation.

Second, the authors fix CSD-CSD dimers, whereas these interactions are expected to be quite dynamic. In the particular example of HP1 proteins, having dimerization equilibria may change the behavior of complex mixtures significantly, e.g. in view of the proposed accumulation of HP1b at a phase boundary. This point would warrant more discussion in the paper. Moreover, the biological plausibility of such a behavior would be interesting. Is there any experimental data supporting such assemblies?

We appreciate the reviewer's insightful comment regarding the dynamic nature of CSD-CSD interactions in HP1 proteins. Our assumption of fixing CSD-CSD dimers is grounded on reported dissociation constant (Kd) values for HP1α and HP1β, which are within the nanomolar range, indicative of strong dimerization affinity (4, 8). While the precise Kd values for HP1γ are not available, a study has demonstrated that HP1γ dimerization is crucial for its interaction with chromatin, suggesting a similar strong dimerization tendency as its paralogs (9, 10). Furthermore, evidence from the literature underscores the dimeric functionality of HP1 paralogs facilitated by their ChromoShadow Domains (CSD), which are instrumental in forming stable genomic domains and engaging in crucial interactions within chromatin architecture (5, 6, 11).

However, we acknowledge that despite the strong dimerization affinity, the CSD-CSD interactions exhibit dynamics, which may influence the behavior of complex mixtures, particularly at phase boundaries. A study by Nielsen et al. (12) shows that mammalian HP1 paralogs can interact directly with one another to form heterodimers. Moreover, the CSD-CSD interface has been shown to act as a hub for transient interactions with diverse binding partner proteins (5, 13). These experimental observations reflect the dynamic nature of CSD-CSD interactions. However, due to the computational constraints and the focus of our study, a simplified static model was employed to gain initial insights into the phase separation behaviors of HP1 paralogs. We believe that the dynamic nature of CSD-CSD interactions and its implications for phase behavior in complex mixtures form an exciting avenue for future computational and experimental studies.

In light of the reviewer’s comment, we have expanded our discussion in the 6th paragraph of the Discussions Section:

“... It is important to emphasize that our model is predicated on the assumption that HP1 proteins establish stable chromoshadow domain (CSD-CSD) dimers, a hypothesis supported by their Kd values being in the nanomolar range (13, 53). While this simplification serves as a useful starting point, it may not fully capture the dynamic nature of HP1 dimerization. Further computational and experimental studies are needed to understand better the behavior of the complex mixtures of HP1 paralogs, particularly at phase boundaries.”

References:

1. R. M. Regy, J. Thompson, Y. C. Kim, J. Mittal, Improved coarse‐grained model for studying sequence dependent phase separation of disordered proteins. Protein Sci., doi: 10.1002/pro.4094 (2021).

2. L. Wang, Y. Gao, X. Zheng, C. Liu, S. Dong, R. Li, G. Zhang, Y. Wei, H. Qu, Y. Li, C. D. Allis, G. Li, H. Li, P. Li, Histone Modifications Regulate Chromatin Compartmentalization by Contributing to a Phase Separation Mechanism. Mol. Cell 76, 646-659.e6 (2019).

3. S. Wohl, M. Jakubowski, W. Zheng, Salt-Dependent Conformational Changes of Intrinsically Disordered Proteins. J. Phys. Chem. Lett. 12, 6684–6691 (2021).

4. C. Her, T. M. Phan, N. Jovic, U. Kapoor, B. E. Ackermann, A. Rizuan, Y. C. Kim, J. Mittal, G. T. Debelouchina, Molecular interactions underlying the phase separation of HP1α: role of phosphorylation, ligand and nucleic acid binding. Nucleic Acids Res., gkac1194 (2022).

5. A. G. Larson, D. Elnatan, M. M. Keenen, M. J. Trnka, J. B. Johnston, A. L. Burlingame, D. A. Agard, S. Redding, G. J. Narlikar, Liquid droplet formation by HP1α suggests a role for phase separation in heterochromatin. Nature 547, 236–240 (2017).

6. M. M. Keenen, D. Brown, L. D. Brennan, R. Renger, H. Khoo, C. R. Carlson, B. Huang, S. W. Grill, G. J. Narlikar, S. Redding, HP1 proteins compact dna into mechanically and positionally stable phase separated domains. eLife 10, 1–38 (2021).

7. W. Qin, A. Stengl, E. Ugur, S. Leidescher, J. Ryan, M. C. Cardoso, H. Leonhardt, HP1β carries an acidic linker domain and requires H3K9me3 for phase separation. Nucleus 12, 44–57 (2021).

8. S. V. Brasher, The structure of mouse HP1 suggests a unique mode of single peptide recognition by the shadow chromo domain dimer. EMBO J. 19, 1587–1597 (2000).

9. X. Li, S. Wang, Y. Xie, H. Jiang, J. Guo, Y. Wang, Z. Peng, M. Hu, M. Wang, J. Wang, Q. Li, Y. Wang, Z. Liu, Deacetylation induced nuclear condensation of HP1γ promotes multiple myeloma drug resistance. Nat. Commun. 14, 1290 (2023).

10. Y. Mishima, C. D. Jayasinghe, K. Lu, J. Otani, M. Shirakawa, T. Kawakami, H. Kimura, H. Hojo, P. Carlton, S. Tajima, I. Suetake, Nucleosome compaction facilitates HP1γ binding to methylated H3K9. Nucleic Acids Res. 43, 10200–10212 (2015).

11. D. O. Trembecka-Lucas, J. W. Dobrucki, A heterochromatin protein 1 (HP1) dimer and a proliferating cell nuclear antigen (PCNA) protein interact in vivo and are parts of a multiprotein complex involved in DNA replication and DNA repair. Cell Cycle 11, 2170–2175 (2012).

12. A. L. Nielsen, M. Oulad-Abdelghani, J. A. Ortiz, E. Remboutsika, P. Chambon, R. Losson, Heterochromatin formation in mammalian cells: Interaction between histones and HP1 Proteins. Mol. Cell 7, 729–739 (2001).

13. A. Thiru, D. Nietlispach, H. R. Mott, M. Okuwaki, D. Lyon, P. R. Nielsen, M. Hirshberg, A. Verreault, N. V. Murzina, E. D. Laue, Structural basis of HP1/PXVXL motif peptide interactions and HP1 localisation to heterochromatin. EMBO J. 23, 489–499 (2004).

14. P. Yu Chew, J. A. Joseph, R. Collepardo-Guevara, A. Reinhardt, Thermodynamic origins of two-component multiphase condensates of proteins. Chem. Sci. 14, 1820–1836 (2023).

**Recommendations for the authors:**
In this important work, the authors apply a residue-resolution protein coarse-grained model to investigate the differences in molecule dimensions and phase behaviour of three HP1 paralogs, HP1 paralog mixtures, and HP1/DNA mixtures. The simulations are well designed to investigate the impact of HP1 sequence on its phase behaviour. The work reveals that electrostatic interactions are a key determinant of HP1 paralog phase behaviour; hence advancing our understanding of the molecular mechanisms driving the phase separation behaviour of HP1 paralogs. Notably, the authors uncovered a nuanced interplay between the folded and disordered domains of HP1. Although disordered regions have traditionally been linked to driving phase separation through their capacity for forming multivalent interactions, the authors demonstrate that the contribution of the CD cannot be overlooked, as it significantly impacts the saturation concentration.Essential revisions (based on reviewers assessment below):1. The manuscript describes the results of both single-molecule simulations and direct coexistence simulations. However, it is not very easy for the reader to determine which types simulations were performed in each section. The details on the simulations input parameters are also missing. Such details are needed throughout, i.e. to allow readers to follow the work and its implications. For instance, the specific salt conditions employed in the simulations are not explicitly stated. Since HP1 charge is presented as a key regulator for the modulation of HP1 paralogs radii of gyration and their phase behaviour, it is crucial for the authors to explicitly describe the salt concentration used for the different simulations and highlight how the relative differences observed are expected to change as the salt concentration decreases/increases.

We have turned the first sentences in the paragraphs into subtitles to describe the results of single homodimers in dilute phase and multi-dimers in phase coexistence simulations.

“Sequence variation affects the conformations of HP1 paralogs in the dilute phase.”

“Sequence variation in HP1 paralogs leads to their distinct phase separation behaviors.”

To improve the clarity, we have also added the following sentence to Fig. 2 caption.

“… Figs. 2a-e show the results obtained under dilute conditions, while Figs. 2f-m illustrate the conditions of phase coexistence.”

We have specified the salt concentration used in the CG simulations in the Materials and Methods section and the relevant figure captions to improve clarity. We also addressed the reviewer’s comment on salt concentration in the public review above.

1. Since direct coexistence simulations suffer from important finite-size effects, especially for multi-component mixtures as those investigated here, describing how many proteins/DNA copies were used per system, the size of the simulation, and which checks were done to check for finite-size effects is important. Regarding this point, estimating C_sat from Direct Coexistence simulations is extremely challenging, given the sensitivity of the dilute phase concentration to the box dimensions. Hence, it would be valuable if the authors clarify that the differences on C_sat provided represent a qualitative comparison and are sensitive to the simulation conditions. Importantly, the observation of spatial segregation of components in multi-component condensates could be an artefact of the box dimensions, relative copies of the various components, and overall system density.

We appreciate the reviewer’s concern regarding the finite-size effects in phase coexistence simulations and potential artifacts arising from box dimensions and system composition. In response to this, we have expanded the Materials and Methods section to elaborate on the specific checks to examine the finite-size effects. The new texts and additional SI figures are shown below.

“Previous studies have demonstrated that slab geometry can help mitigate finite-size effects and facilitate efficient sampling of the phase diagram (41). To assess the potential impact of finite-size effects with our chosen box dimensions, we conducted a test using the HP1α homodimer, which serves as a representative system given the comparable sequence lengths of HP1 paralogs and their chimeras. By reducing the system size by 30% and constructing its phase diagram, we observed that both the original system size (50 dimers) and the reduced counterpart (35 dimers) produced similar phase diagrams, with critical temperatures of 353.3 K and 352.1 K, respectively, as shown in Figs. S4a,b.

We further evaluated the influence of the xy cross-sectional area on the measurement of Csat. With the z-direction box length fixed at 1190 Å³, we varied the xy cross-sectional areas (120x120, 150x150, and 200x200 Å²) while maintaining the protein density consistent with the control case (170x170 Å²). Given that HP1 dimers are multidomain proteins, a 120x120 Å² cross-section was the minimum size feasible to prevent particle overlap in HOOMD simulations due to the constraints of the small box size. Our findings indicate that the condensates remained stable across all tested cross-sectional areas and that there were no significant differences in Csat measurements within the margin of error, as depicted in Figs. S4c,d. These results confirm that our chosen box size is sufficiently large to minimize finite-size effects, thus ensuring the robustness of our results.”

**Author response image 3. sa3fig3:** Finite-size analysis. (a) Phase diagrams for the HP1α homodimer (50 dimers) and for a system reduced in size by 30% (35 dimers), with critical temperatures of 353.3 K and 352.1 K, respectively. (b) Density profiles of HP1α and its reduced size counterpart at various temperatures. (c, d) Density profiles and snapshots of HP1α homodimer simulation with box dimensions of 170x170x1190 Å3 and for systems with z-direction length fixed at 1190 Å and varying cross-sectional areas: 120x120, 150x150, and 200x200 Å2. The black dashed line shows the simulated saturation concentration of wildtype HP1α homodimer in the box dimensions of 170x170x1190 Å3. The simulations were conducted at 320 K and 100 mM salt concentrations. The error bars represent the standard deviation from triplicate simulation sets.

In response to the observed spatial segregation in our multi-component condensates, we have carefully considered finite-size effects and are confident that the segregation reflects genuine phase behavior rather than an artifact of simulation parameters. This interpretation is supported by findings from Chew et al. (14), who observed similar multilayered condensates and conducted thorough validations to verify these phases. To clarify our approach, we have included additional details in the Materials and Methods section of our manuscript.

“... By selecting a box size that minimizes finite-size effects, we can ensure that the spatial segregation observed in our multi-component condensates reflects genuine phase behavior. This finding aligns with Chew et al. (66), who also reported well-separated multilayered condensates and conducted thorough validations to confirm these phases.”

1. The authors should provide a clearer assessment of the quantitative precision of their model. For instance, the authors use all-atom simulations to compare with CG interaction maps. The all-atom maps are sparser due to less sampling, but the authors state that the maps are 'in good agreement'. How do the authors judge this? The issue of model validation is very important: to illustrate, HP1a is anticipated to undergo phase separation primarily under low salt concentrations. Does the model effectively capture this sensitivity to salt conditions? While numerous findings in the manuscript likely remain valid, it could be beneficial to acknowledge potential limitations tied to the simulations. For instance, might the absence of quantitative precision impact certain predictions, such as the CD's influence on phase separation?

This is similar to the point made by Reviewer 2 in the Public Review. We have addressed these questions in the public review and incorporated new plots (Fig. S1 f-h) in the SI.

1. The authors fix CSD-CSD dimers, whereas these interactions are expected to be quite dynamic. In the particular example of HP1 proteins, having dimerization equilibria may change the behaviour of complex mixtures significantly, e.g. in view of the proposed accumulation of HP1b at a phase boundary. This point warrants more discussion in the paper.

We have addressed the comment in the public review and extended the discussion in the Discussion section.

**Reviewer #2 (Recommendations For The Authors):**
The authors use all-atom simulations to validate their CG model. In Figure S1, they compare interaction maps. Of course, the AA maps are sparser due to less sampling, but the authors state that the maps are 'in good agreement'. How do the authors judge this (they do not look very similar to me, e.g. the NTD-hinge interactions are mostly lacking)?

This is similar to Reviewer 1’s concern. We agree that the AA simulations are moderately limited over 5 μs due to the large size of the HP1 proteins (~400 residues in a dimer). However, the expansion trends of the average dimensions of the HP1 paralogs agree with the CG simulations (Fig. S1 a,b). Regarding the AA contact maps, we agree that they are relatively sparse, which makes it difficult to compare them to the CG maps. We have added new plots (Fig. S1 f-h) to show the correlation of the vdW contact probability per residue for each paralog in the AA and CG simulations. The Pearson correlation coefficients are approximately 0.86, suggesting a strong positive linear correlation in the contact propensity between AA and CG simulations.